# Provably Bounding Neural Network Preimages

**Suhas Kotha**[*]
Carnegie Mellon
suhask@andrew.cmu.edu

**Christopher Brix**[*]
RWTH Aachen
brix@cs.rwth-aachen.de

**Zico Kolter**
Carnegie Mellon
Bosch Center for AI
zkolter@cs.cmu.edu

**Krishnamurthy (Dj) Dvijotham**[†]
Google DeepMind
dvijothamcs@gmail.com

**Huan Zhang**[†]
UIUC
huan@huan-zhang.com

## Abstract

Most work on the formal verification of neural networks has focused on bounding the set of outputs that correspond to a given set of inputs (for example, bounded perturbations of a nominal input). However, many use cases of neural network verification require solving the inverse problem, or over-approximating the set of inputs that lead to certain outputs. We present the INVPROP algorithm for verifying properties over the preimage of a linearly constrained output set, which can be combined with branch-and-bound to increase precision. Contrary to other approaches, our efficient algorithm is GPU-accelerated and does not require a linear programming solver. We demonstrate our algorithm for identifying safe control regions for a dynamical system via backward reachability analysis, verifying adversarial robustness, and detecting out-of-distribution inputs to a neural network. Our results show that in certain settings, we find over-approximations over $2500\times$ tighter than prior work while being $2.5\times$ faster. By strengthening robustness verification with output constraints, we consistently verify more properties than the previous state-of-the-art on multiple benchmarks, including a large model with 167k neurons in VNN-COMP 2023. Our algorithm has been incorporated into the $\alpha,\beta$-CROWN verifier, available at https://abcrown.org.

## 1   Introduction

Applying neural networks to safety-critical settings often requires reasoning about constraints on the inputs and outputs of a neural network. For example, for a physical system controlled by a neural network policy, it is of interest to understand which initial states will lead to an unsafe state such as colliding with an obstacle. Formal verification of neural networks seeks to provide provable guarantees demonstrating that networks satisfy formal specifications on their inputs and outputs. Most work to date has focused on developing algorithms that can bound the outputs of a neural network given constraints on the inputs, which can be used for applications such as analyzing the robustness of a neural network to perturbations of a given input [Wong and Kolter, 2018, Dvijotham et al., 2018, Zhang et al., 2018a, Raghunathan et al., 2018, Gehr et al., 2018].

In this work, we address the inverse problem of over-approximating a neural network's preimage: given a set of outputs $\mathcal{S}_{\text{out}}$ (described by linear constraints on the network output), we seek to find a set that provably contains all inputs that lead to such outputs. For example, for our neural network policy, this would correspond to the states that collide with an obstacle one step in the future. Though the verification problem is already challenging due to non-convexity and high dimensionality, this new problem is even more difficult since neural networks are generally not invertible.

---

* Equal contribution, † Equal advising

Instructions for reproducing our results are available at https://github.com/kothasuhas/verify-input

Specifically, representative efficient verifiers (such as state-of-the-art bound-propagation-based methods [Zhang et al., 2022]) can only compute bounds utilizing constraints on the input and critically depend on having tight bounds on intermediate activations. In the setting of this paper, however, the bounds derived from the input constraints are almost vacuous, since the only constraints on the input are that it should be from the valid input domain. We efficiently solve this problem by significantly generalizing the existing bound propagation-based verification framework, allowing one to leverage output constraints when tightening the intermediate activations. Our contributions are as follows:

• We develop an effective bound propagation framework, Inverse Propagation for Neural Network Verification (INVPROP), for the *inverse verification problem* for neural networks, i.e., the problem of over-approximating the set of inputs that leads to a given set of outputs. Importantly, INVPROP requires no linear programming solver and can compute bounds on any intermediate layer.

• We unify INVPROP and traditional bound propagation into a more general verification framework, allowing us to connect our method to standard tools, such as the state-of-the-art bound propagation tool $\alpha,\beta$-CROWN [Zhang et al., 2018a, Xu et al., 2021, Wang et al., 2021, Zhang et al., 2022, Brix et al., 2023, Müller et al., 2023]. Our contribution allows $\alpha,\beta$-CROWN to tighten intermediate bounds with respect to output constraints, which could not be done by the original tool.

• We demonstrate that tight inverse verification requires multiple iterative refinements of intermediate bounds. While layer bounds in standard bound propagation only depend on the bounds of their predecessors, INVPROP incorporates bounds of *all* layers in the network.

• We improve the state of the art on a control benchmark [Rober et al., 2022a,b] by providing $2500\times$ tighter bounds, $2.5\times$ faster, for a Double Integrator and $257\times$ tighter bounds, $3.29\times$ faster, for a 6D Quadrotor. Furthermore, we demonstrate that INVPROP can strengthen robustness verification with output constraints and verify more robustness properties in less time compared to existing tools. Finally, we demonstrate its applicability in OOD detection.

## 2 Background and Problem Setup

### 2.1 Notation

We use $[L]$ for $L \in \mathbb{N}$ to refer to the set $\{1, 2, \dots, L\}$, $\mathbf{W}^{(i)}_{:,j}$ to refer to column $j$ of the matrix $\mathbf{W}^{(i)}$, $[\cdot]_+$ to refer to $\max(0, \cdot)$, and $[\cdot]_-$ to refer to $-\min(0, \cdot)$. We use boldface symbols for vectors and matrices (such as $\boldsymbol{x}^{(i)}$ and $\mathbf{W}^{(i)}$) and regular symbols for scalars (such as $x^{(i)}_j$). We use $\boldsymbol{x} \odot \boldsymbol{y}$ to denote element-wise multiplication of vectors $\boldsymbol{x}, \boldsymbol{y}$.

We define an $L$ layer ReLU neural network by its weight matrices $\mathbf{W}^{(i)}$ and bias vectors $\mathbf{b}^{(i)}$ for $i \in [L]$. The output of the neural network for the input $\hat{\boldsymbol{x}}^{(0)}$ from a bounded input domain $\mathcal{X}$ is computed by alternately applying linear layers $\boldsymbol{x}^{(i)} = \mathbf{W}^{(i)}\hat{\boldsymbol{x}}^{(i-1)} + \mathbf{b}^{(i)}$ and ReLU layers $\hat{\boldsymbol{x}}^{(i)} = \max(\mathbf{0}, \boldsymbol{x}^{(i)})$ until we receive the output $\boldsymbol{x}^{(L)}$ (which we refer to as the logits). Note that we treat softmax as a component of the loss function, not the neural network.

### 2.2 Problem Statement

Given a neural network $f : \mathcal{X} \subseteq \mathbb{R}^{\text{in}} \to \mathbb{R}^{\text{out}}$ and an output constraint $\mathcal{S}_{\text{out}} \subseteq \mathbb{R}^{\text{out}}$, we want to compute $f^{-1}(\mathcal{S}_{\text{out}}) \subseteq \mathcal{X}$. Since precisely computing or expressing $f^{-1}(\mathcal{S}_{\text{out}})$ is an intractable problem in general, we strive to compute a tight over-approximation $\mathcal{S}_{\text{over}}$ such that $f^{-1}(\mathcal{S}_{\text{out}}) \subseteq \mathcal{S}_{\text{over}}$. In particular, we target the convex hull of the preimage via a cutting-plane representation. $\mathcal{S}_{\text{out}}$ will be defined by a set of linear constraints parameterized by $\mathbf{H}f(\boldsymbol{x}) + \mathbf{d} \le \mathbf{0}$ in this work.

### 2.3 Applications

**Backward Reachability Analysis for Neural Feedback Loops.** Establishing safety guarantees for neural network policies is a challenging task. One problem of interest is to find a set of initial states that does not reach a particular set of future states under the neural network policy. This can be helpful in collision avoidance or control with safety constraints. For example, consider a discrete-time double integrator controller [Hu et al., 2020] where the state at time $t + 1$ can be directly computed based on state at time $t$ following the equation

$$\boldsymbol{x}_{t+1} = f(\boldsymbol{x}_t) = \begin{bmatrix} 1 & 1 \\ 0 & 1 \end{bmatrix} \boldsymbol{x}_t + \begin{bmatrix} 0.5 \\ 1 \end{bmatrix} \pi(\boldsymbol{x}_t)$$

with policy $\pi : \mathbb{R}^2 \to \mathbb{R}$. If there is an obstacle in the room covering the region $[4.5, 5.0] \times [-0.25, 0.25]$, it is of interest to understand which states will enter the unsafe region in the next time-step. We can represent this obstacle set with linear constraints that define $\mathcal{S}_{\text{out}}$.

$f^{-1}(\mathcal{S}_{\text{out}})$ denotes the set of states $\boldsymbol{x_t}$ such that $\boldsymbol{x_{t+1}}$ lies in the unsafe region given the control policy $\pi$. Overapproximating this set allows us to define the set of states to avoid one timestep in advance. We can compose $f^{-1}$ with itself $t$ times to obtain the set of initial states where $\pi$ would drive the system to the unsafe set after $t$ steps.

**Robustness Verification**  Adversarial robustness is one classic problem for neural network verification where verifiers assess whether all perturbations of an input yield the same classification. INVPROP can be used to speed up this verification by exploiting an implicit output constraint. We follow Wang et al. [2021] and use the canonical form of verifying a property by proving that $\min_{x \in \mathcal{X}} f(x) > 0$. $f(x)$ will be negative if and only if $x$ is an adversarial example. This minimization problem can be rewritten as $\inf_{x \in X \wedge f(x) \leq 0} f(x)$ [Yang et al., 2021b]. If no adversarial example exists, the infimum is computed over an empty set and returns positive infinity, indicating local robustness. If at least one adversarial example exists, the result is guaranteed to be negative. With the output constraint $f(x) \geq 0$, we only analyze inputs that yield an incorrect classification, reducing the search space and tightening verification bounds.

**Determining what inputs lead to confident predictions.**  It is challenging to train a classifier that knows when it doesn't know. A simple and effective approach is to train with an additional logit representing OOD abstention. Labeled data for this additional class is generated via outlier exposure, or adding synthetic training data known to be OOD [Hendrycks et al., 2018, Chen et al., 2021].

Consider the logits $y$ produced by a binary classifier trained in this manner. The classifier can classify in-distribution data by comparing $y_0$ and $y_1$, the logits corresponding to the two in-distribution labels. The quantity $\max(y_0, y_1) - y_2$, known as the logit gap, can be used to identify out-of-distribution data (Fig. 7 in Appendix A). Can the logit gap correctly identify data far from the training data as being OOD? To answer this quesiton, we can compute the preimage of the set

$$\mathcal{S}_{\text{out}} = \{\boldsymbol{y} : \max(y_0, y_1) \geq y_2\}$$

We detail how to model this set with linear constraints in Appendix F.1. INVPROP enables us to over-approximate $f^{-1}(\mathcal{S}_{\text{out}})$, answering whether the classifier learned to correctly identify OOD data.

## 3  Approach

### 3.1  Convex over-approximation

Suppose we want to find a convex over-approximation of $f^{-1}(\mathcal{S}_{\text{out}})$. The tightest such set is its convex hull, which is the intersection of all half-spaces that contain $f^{-1}(\mathcal{S}_{\text{out}})$ [Boyd et al., 2004]. This intersection is equivalent to $\bigcap_{\boldsymbol{c} \in \mathbb{R}^{\text{in}}} \{\boldsymbol{x} : \boldsymbol{c}^\top \boldsymbol{x} \geq \min_{f(\boldsymbol{x}') \in \mathcal{S}_{\text{out}}} \boldsymbol{c}^\top \boldsymbol{x}'\}$, which means that we can build an over-approximation by taking this intersection for finitely many $\boldsymbol{c}$. Furthermore, replacing the minimization problem with a lower bound to its value still yields a valid half-space, where a tighter bound yields a tighter approximation.

We focus on convex over-approximations for two reasons: First, checking that the preimage satisfies a linear constraint $\boldsymbol{c}^\top \boldsymbol{x} + d \geq 0$ is equivalent to minimizing the linear function $\boldsymbol{x} \mapsto \boldsymbol{c}^\top \boldsymbol{x}$ over the preimage, which is in turn equivalent to minimizing the linear function over $\text{CONV}\left(f^{-1}(\mathcal{S}_{\text{out}})\right)$ [Boyd et al., 2004]. Second, the convex hull and its tractable over-approximations are conveniently represented as intersections of linear constraints on the preimage, each of which can be quickly computed after a single precomputation phase to tighten bounds on intermediate neurons using the INVPROP algorithm outlined in the next section.

For the above reasons, we will focus on solving the following constrained optimization problem.

$$\min_{\boldsymbol{x}} \quad \boldsymbol{c}^\top \boldsymbol{x}; \qquad \text{s.t. } \boldsymbol{x} \in \mathcal{X}; \quad f(\boldsymbol{x}) \in \mathcal{S}_{\text{out}} \tag{1}$$

Note that this differs from the widely studied forward verification problem, which can be phrased as

$$\min_{\boldsymbol{x} \in \mathcal{S}_{\text{in}}} \quad \boldsymbol{c}^\top f(\boldsymbol{x})$$

where $\mathcal{S}_{\text{in}}$ is a set representing constraints on the input, and the goal of the verification is to establish that the output $f(\boldsymbol{x})$ satisfies a linear constraint of the form $\boldsymbol{c}^\top f(\boldsymbol{x}) \geq d$ for all $x \in \mathcal{S}_{\text{in}}$.

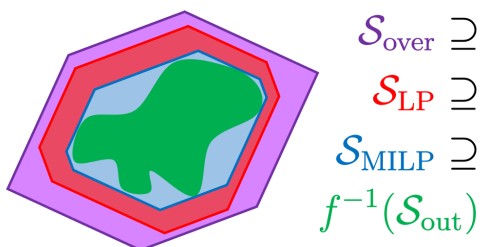

$\mathcal{S}_{\text{over}} \supseteq$

$\mathcal{S}_{\text{LP}} \supseteq$

$\mathcal{S}_{\text{MILP}} \supseteq$

$f^{-1}(\mathcal{S}_{\text{out}})$

Figure 1: **Visualization of relaxations.** The inner green region depicts the true $f^{-1}(\mathcal{S}_{\text{out}})$, the blue relaxation depicts the intersection of finite half-spaces solved via MILP, the red relaxation displays the same via LP, and the purple relaxation displays the same via INVPROP. Though this diagram displays looseness, we provide a comprehensive methodology to reduce the error in all three relaxations up to arbitrary precision (Section 3.3).

## 3.2 The INVPROP Algorithm

As a brief overview of our method, we first construct the Mixed Integer Linear Program (MILP) for solving this optimization problem, which generates the over-approximation $\mathcal{S}_{\text{MILP}}$ when solved for finitely many $\boldsymbol{c}$. Then, we relax the MILP to a Linear Program (LP), which can construct the over-approximation $\mathcal{S}_{\text{LP}}$. Finally, we relax the LP via its Lagrangian dual, which will be used to construct our over-approximation $\mathcal{S}_{\text{over}}$. This chain of relaxations is visualized in Figure 1.

**The Mixed Integer Linear Programming (MILP) Formulation**   For feed-forward ReLU neural networks, the non-linearities from the max operator can be encoded via integer variables, and this problem admits a MILP encoding similar to prior work in adversarial robustness [Tjeng et al., 2017]. Problem (1) is equivalent to:

$$\min_{\boldsymbol{x},\hat{\boldsymbol{x}}} \quad \boldsymbol{c}^\top \boldsymbol{x} \tag{2a}$$

$$\text{s.t.} \quad \boldsymbol{x} \in \mathcal{X}; \quad \hat{\boldsymbol{x}}^{(0)} = \boldsymbol{x}; \quad \mathbf{H}\boldsymbol{x}^{(L)} + \mathbf{d} \le \mathbf{0}; \tag{2b}$$

$$\boldsymbol{x}^{(i)} = \mathbf{W}^{(i)}\hat{\boldsymbol{x}}^{(i-1)} + \mathbf{b}^{(i)} \quad i \in [L]; \tag{2c}$$

$$\hat{\boldsymbol{x}}^{(i)} = \max(0, \boldsymbol{x}^{(i)}); \quad i \in [L-1] \tag{2d}$$

**The Linear Programming (LP) Formulation**   Unfortunately, finding an exact solution to this MILP is NP-complete [Katz et al., 2017]. To sidestep the intractability of exact verification, we can compute lower bounds for this program via its convex relaxation. We consider bounds on the outputs of intermediate layers:

$$l_j^{(i)} \le x_j^{(i)} \le u_j^{(i)}$$

Based on these bounds, we can take the ReLU triangle relaxation [Ehlers, 2017] to get the LP

$$\min_{\boldsymbol{x},\hat{\boldsymbol{x}}} \quad \boldsymbol{c}^\top \boldsymbol{x} \tag{3a}$$

$$\text{s.t.} \quad \boldsymbol{x} \in \mathcal{X}; \quad \boldsymbol{l}^{(0)} \le \boldsymbol{x} \le \boldsymbol{u}^{(0)}; \quad \hat{\boldsymbol{x}}^{(0)} = \boldsymbol{x}; \quad \mathbf{H}\boldsymbol{x}^{(L)} + \mathbf{d} \le \mathbf{0} \tag{3b}$$

$$\boldsymbol{x}^{(i)} = \mathbf{W}^{(i)}\hat{\boldsymbol{x}}^{(i-1)} + \mathbf{b}^{(i)} \tag{3c}$$

$$\mathbf{0} \le \hat{\boldsymbol{x}}^{(i)}; \quad \boldsymbol{x}^{(i)} \le \hat{\boldsymbol{x}}^{(i)}; \quad \hat{\boldsymbol{x}}^{(i)} \le \frac{\boldsymbol{u}^{(i)}}{\boldsymbol{u}^{(i)} - \boldsymbol{l}^{(i)}} \odot \left( \boldsymbol{x}^{(i)} - \boldsymbol{l}^{(i)} \right); \quad i \in [L] \tag{3d}$$

where the bounds $\boldsymbol{l}^{(0)} \le \boldsymbol{x} \le \boldsymbol{u}^{(0)}$ are either the bounds implicit in $\mathcal{X}$, or a refinement of these obtained via previous rounds of bound propagation or input branching (see Algorithm 1 for details). Most efficient neural network verifiers do not solve an LP formulation of verification directly because LP solvers are often slow for problem instances involving neural networks. Instead, the bound propagation framework [Zhang et al., 2018a, Wang et al., 2021, Zhang et al., 2022] is a practical and efficient way to lower bound the LP relaxation of the forward verification problem. However, there are *two major roadblocks* to applying existing methods here: Typical bound-propagation cannot directly handle the output constraints (Eq. 3b) and the objective involving input layer variables (Eq. 3a). This is true for optimizing bounds on the input, intermediate layers, and the output layer, all of which need to be iteratively tightened as demonstrated later.

**The Inverse Propagation (INVPROP) Formulation**    By changing the LP above to optimize the quantity $\hat{x}_j^{(0)}$ (input layer) or $x_j^{(i)}$ (intermediate layers) for $i \in [L-1]$, the bounds for the $j$-th neuron of layer $i$ can be tightened in separate LP calls. However, this program is too expensive to be run multiple times for each neuron.

Inspired by the success of CROWN-family neural network verifiers [Zhang et al., 2018a, Xu et al., 2021, Wang et al., 2021, Zhang et al., 2022], we efficiently lower bound the solution of the LP by optimizing its Lagrangian dual. This dual is highly structured [Wong and Kolter, 2018], allowing us to bound input half-spaces $\boldsymbol{c}^\top \boldsymbol{x}$ and intermediate bounds $l_j^{(i)}, u_j^{(i)}$ by closed-form expressions of the dual variables. Our main generalization is the ability to optimize input or intermediate layer bounds with the output constraints **after** them in the neural network, as shown in the following theorem.

**Theorem 1** (Bounding input half-spaces). *Given an output set $\mathcal{S}_{out} = \{\boldsymbol{y} : \mathbf{H}\boldsymbol{y} + \mathbf{d} \leq \mathbf{0}\}$ and vector $\boldsymbol{c}$, $g_{\boldsymbol{c}}(\boldsymbol{\alpha}, \boldsymbol{\gamma})$ is a lower bound to the linear program in (3) for $\mathbf{0} \leq \boldsymbol{\alpha} \leq \mathbf{1}$, $\boldsymbol{\gamma} \geq \mathbf{0}$, and $g_{\boldsymbol{c}}$ defined via*

$$g_{\boldsymbol{c}}(\boldsymbol{\alpha}, \boldsymbol{\gamma}) = \left[ \boldsymbol{c}^\top - \boldsymbol{\nu}^{(1)\top} \mathbf{W}^{(1)} \right]_+ \boldsymbol{l}^{(0)} - \left[ \boldsymbol{c}^\top - \boldsymbol{\nu}^{(1)\top} \mathbf{W}^{(1)} \right]_- \boldsymbol{u}^{(0)}$$

$$- \sum_{i=1}^{L} \boldsymbol{\nu}^{(i)\top} \mathbf{b}^{(i)} + \sum_{i=1}^{L-1} \sum_{j \in \mathcal{I}^{\pm(i)}} \left[ \frac{u_j^{(i)} l_j^{(i)} [\hat{\nu}_j^{(i)}]_+}{u_i^{(j)} - l_i^{(j)}} \right]$$

*where every term can be directly recursively computed via*

$$\mathcal{I}^{-(i)} = \{j : u_j^{(i)} \leq 0\}; \quad \mathcal{I}^{+(i)} = \{j : l_j^{(i)} \geq 0\}; \quad \mathcal{I}^{\pm(i)} = \{j : l_j^{(i)} < 0 < u_j^{(i)}\}$$

$$\boldsymbol{\nu}^{(L)} = -\boldsymbol{\gamma}; \quad \hat{\nu}_j^{(i)} = \boldsymbol{\nu}^{(i+1)\top} \mathbf{W}_{:,j}^{(i+1)} \quad \forall i \in [L-1]$$

$$\nu_j^{(i)} = \begin{cases} \hat{\nu}_j^{(i)}, & j \in \mathcal{I}^{+(i)} \\ 0, & j \in \mathcal{I}^{-(i)} \\ \frac{u_j^{(i)}}{u_i^{(j)} - l_i^{(j)}} [\hat{\nu}_j^{(i)}]_+ - \alpha_j^{(i)} [\hat{\nu}_j^{(i)}]_-, & j \in \mathcal{I}^{\pm(i)} \end{cases} \quad \forall i \in [L-1]$$

*Proof.* Full proof is presented in Appendix D.  □

In Appendix C, we show how to bound intermediate neurons in the network using a similar approach. Since the intermediate bounds might depend on bounds on neurons in subsequent layers (due to the output constraint), we cannot simply optimize bounds in a single forward pass layer by layer, unlike prior work such as $\alpha$-CROWN [Xu et al., 2021]. Instead, we must iteratively tighten intermediate layer bounds with respect to the tightest bounds on all neurons computed thus far. This iterative approach can tighten the initially loose bounds by several orders of magnitude, as shown in Figure 2. After performing this procedure once, the intermediate bounds can be used to tightly lower bound $\boldsymbol{c}^\top \boldsymbol{x}$ for any $\boldsymbol{c}$ via Theorem 1. Therefore, this computation can be shared across all the constraints $\boldsymbol{c}$ we use to describe $\mathcal{S}_{\text{over}}$. Our algorithm can be expressed in terms of forward/backward passes through layers of the neural network and implemented via standard deep learning modules in libraries like PyTorch [Paszke et al., 2019]. Since all of the operations are auto-differentiable, we can tighten our lower bound using standard gradient ascent (projected by the dual variable constraints).

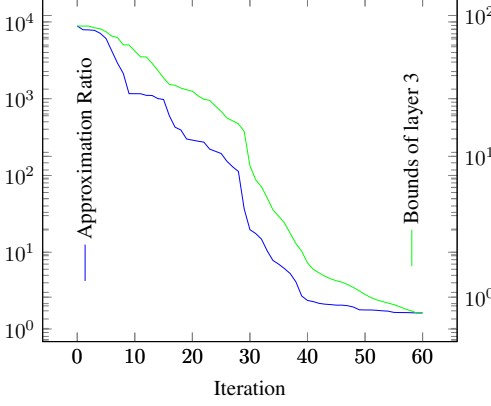

Figure 2: **Necessity of Iterative Tightening.** Our approach enables us to iteratively tighten the bounds of all layers, with each iteration allowing for a smaller approximation ratio with respect to the true preimage. Green: Sum of bound intervals for all neurons in the third layer (second hidden layer); Blue: ratio between volumes of over-approximation and preimage. Measured for step $t = 10$ of the control benchmark defined in Section 4.1. Note that the improvement from iterative tightening is two orders of magnitude for intermediate bounds, and four orders of magnitude for the volume of the over-approximation.

**Algorithm 1** INVPROP. Can be applied to branches for non-convex overapproximations. Lower and upper bound of all neurons and all constraints are optimized using distinct $\alpha$ and $\gamma$ values in $g$.

---

Initialize $\boldsymbol{l}^{(i)}, \boldsymbol{u}^{(i)}$ via cheap bound propagation methods for the forward verification problem
**while** lower bounds for $\boldsymbol{c}^\top \hat{\boldsymbol{x}}^{(0)}$ are improving **do**
    **for** $i \in \{L-1, L-2, \ldots, 1, 0\}$ **do**
        **for** $j \in$ layer $i$ neurons, $b \in \{$lower, upper$\}$ **do**
            Optimize $g_{ij}^b$ from Theorem 2 via gradient ascent to improve bound on neuron $j$ in layer $i$ with sense $b$ (lower/upper)
            **if** $b =$upper **then** update $u_j^{(i)}$ **else** update $l_j^{(i)}$
    Optimize $g_{\boldsymbol{c}}$ from Theorem 2 via gradient ascent to improve lower bound on $\boldsymbol{c}^\top \boldsymbol{x}$ for all $\boldsymbol{c}$

---

**Connection to forward verification** Our bound in Theorem 1 introduces the dual variable $\boldsymbol{\gamma}$, which enforces the output constraint during optimization. In fact, we can use this variable to get a better conceptual interpretation of our result. For the optimization problem in (1), taking the dual with respect to the constraint $\mathbf{H}f(\boldsymbol{x}) + \mathbf{d} \leq \mathbf{0}$ yields the lower bound

$$\max_{\boldsymbol{\gamma}} \min_{\boldsymbol{x}} \quad \boldsymbol{c}^\top \boldsymbol{x} + \boldsymbol{\gamma}^\top \left( \mathbf{H}f(\boldsymbol{x}) + \mathbf{d} \right)$$
$$\text{s.t.} \quad \boldsymbol{x} \in \mathcal{X}; \quad \boldsymbol{\gamma} \geq 0$$

The objective can be represented as minimizing a linear function of the output of a residual neural network with a skip connection from the input to the output layer, subject to constraints on the input. Now, $f(\boldsymbol{x})$ appears in the objective, similar to the standard forward verification problem with an augmented network architecture. Similarly, optimizing an intermediate bound is equivalent to a skip connection from layer $i$ to the output (Appendix E). These connections allow us to implement our method using standard verification tools [Brix et al., 2023, Xu et al., 2021, 2020].

When $f$ is a feedforward ReLU network, the lower bound described in this section is precisely the same as Theorem 1 (since both are solving the dual of the same linear program). This shows that the introduction of $\boldsymbol{\gamma}$ constitutes our generalization to the bound propagation framework.

**Selection of $\boldsymbol{c}$** The dependence of the optimization on $\boldsymbol{c}$ is not a major limitation. The bounds of all intermediate neurons are optimized only with respect to the input and output constraints, not the hyperplanes used to describe the input half-space. Thus, their optimized bounds can be shared across the optimization of all input hyperplanes. For applications such as adversarial robustness, the box constraint of two hyperplanes per dimension could be chosen to scalably bound the input. The selection of appropriate $\boldsymbol{c}$ depends on the application and is not our main focus.

### 3.3 Branch and Bound

Though more rounds of iterative tightening will lower the gap in $\mathcal{S}_{\text{over}} \supseteq \mathcal{S}_{\text{LP}}$, our current formulation still faces two sources of looseness: the gap in $\mathcal{S}_{\text{LP}} \supseteq \mathcal{S}_{\text{MILP}}$ and the gap in $\mathcal{S}_{\text{MILP}} \supseteq f^{-1}(\mathcal{S}_{\text{out}})$. To overcome both of these issues, we can make use of branching [Bunel et al., 2020]. While there are several possibilities here, we focus on input branching, which gave the biggest empirical gains in our experiments described in Section 4. More concretely, we divide the input domain $\mathcal{X} = [\boldsymbol{l}^{(0)}, \boldsymbol{u}^{(0)}]$ into several regions by choosing a coordinate $i$ and computing

$$\mathcal{X}_a = \mathcal{X} \cap \{\boldsymbol{x} : \boldsymbol{x}_i \geq s_i\}, \mathcal{X}_b = \mathcal{X} \cap \{\boldsymbol{x} : \boldsymbol{x}_i \leq s_i\}$$

so that $\mathcal{X} = \mathcal{X}_a \cup \mathcal{X}_b$. Doing this recursively, we obtain several regions and can compute an overapproximation of $f^{-1}(\mathcal{S}_{\text{out}})$ when the input is in each of those regions, and take a union of the resulting sets. The finer the input domain of our over-approximation, the tighter the approximation.

## 4 Results

We evaluate INVPROP on three different benchmarks. Across benchmarks, we find orders of magnitude improvement over prior work, and our methods work even for relatively large networks (167k neurons) and high dimensionality inputs (8112 dimensions). All implementation details are described in Appendix F and the utilized hardware is described in Appendix G.

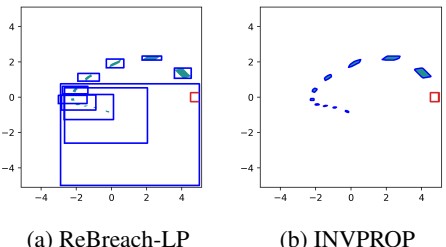

(a) ReBreach-LP     (b) INVPROP

Figure 3: **Double Integrator.** Each green region represents the preimage of the system one to ten steps back from the obstacle (red box). Each preimage is approximated 1 million samples. The blue bounding boxes represent over-approximations, and INVPROP tightly over-approximates preimages even ten steps back.

Table 1: **Double Integrator Results.** IN-VPROP vs SOTA (from Figure 3). We are significantly tighter and faster than ReBreach-LP method even without input branching.

| METHOD | REBREACH-LP | INVPROP |
|---|---|---|
| INPUT BRANCHING | YES | NO |
| APPROX RATIO $\downarrow$ | 4021.47 | 1.46 |
| TIME (SEC) | $42.86 \pm 0.04$ | $17.89 \pm 0.03$ |

## 4.1 Backward Reachability Analysis for Neural Feedback Loops

**Double Integrator** After combining the state-feedback control policy $\pi$ of the benchmark with the state transition function from Section 2.3, (see Section F.2), we get a three layer MLP with 12, 7, and 2 neurons, which is typical for applications in this setting. We model multiple time steps via composing copies of this function.[1] Moreover, we leverage intermediate bounds for time step $t$ to bound the $t + 1$ time step transition. We measure the tightness of an over-approximation using its Approximation Ratio, defined as $\frac{\text{vol}(\mathcal{S}_{\text{over}})}{\text{vol}(f^{-1}(\mathcal{S}_{\text{out}}))}$. Both of these volumes are heuristically estimated with 1 million samples of the input space. While INVPROP allows the optimization of arbitrary input cutting planes, we use 40 planes with slopes of equally distributed angles.

We find that INVPROP is significantly tighter and faster than ReBreach-LP, the SOTA available method for this problem [Rober et al., 2022a,b].[2] As evident in Figure 3, ReBreach-LP suffers from increasingly weak bounds as $t$ increases, whereas our approach is able to compute tight bounds for all $t$. Different to their implementation, we are able to leverage the output constraint to improve the intermediate neuron bounds. Furthermore, we can iteratively tighten these bounds with respect to each other. Even though we optimize many more quantities, our efficient bound propagation allows us to solve the problem faster overall. We provide a visualization of the improvement of the approximation ratio score and intermediate bounds over time in Figure 2. Both our increased tightness and speed are quantified in Table 1.

**6D Quadrotor** We consider over-approximating the preimage of the linearized quadrotor discussed in Figure 15 of Rober et al. [2022b]. This quadrotor has a 6-dimensional state space with 3 dimensions dedicated to position and is given by the dynamics

$$\boldsymbol{x}_{t+1} = \begin{bmatrix} 1 & 0 & 0 & 1 & 0 & 0 \\ 0 & 1 & 0 & 0 & 1 & 0 \\ 0 & 0 & 1 & 0 & 0 & 1 \\ 0 & 0 & 0 & 1 & 0 & 0 \\ 0 & 0 & 0 & 0 & 1 & 0 \\ 0 & 0 & 0 & 0 & 0 & 1 \end{bmatrix} \boldsymbol{x}_t + \begin{bmatrix} 0.5 & 0 & 0 \\ 0 & 0.5 & 0 \\ 0 & 0 & 0.5 \\ 1 & 0 & 0 \\ 0 & 1 & 0 \\ 0 & 0 & 1 \end{bmatrix} \pi(\boldsymbol{x}_t)$$

The policy, after performing the same encoding as the double integrator and clipping policy outputs, forms a five layer MLP with 26, 26, 9, 9, and 3 neurons. For these dynamics, we want to prove that the initial state range of $[-5.25, -4.75] \times [-.25, .25] \times [2.25, 2.75] \times [0.95, 0.99] \times [-0.01, 0.01] \times [-0.01, 0.01]$ (black box in Figure 4) will never collide with the obstacle at $[-1, 1] \times [-1, 1] \times [1.5, 3.5] \times [-1, 1] \times [-1, 1] \times [-1, 1]$ (red box). We try to find the tightest bounding box in the 3 position dimensions, reflecting a total of 6 input cutting planes (one for each direction in each dimension). To benchmark against prior work, we run ReBreach-LP with no output partitioning and

---

[1]For example, the 10 time step transition function can be represented as a 21 layer MLP (after fusing consecutive linear layers in the composition).

[2]We do not compare with Everett et al. [2022] which improves upon ReBreach-LP, as their implementation faces numerical instability when applied for ten timesteps. However, preliminary experiments show that we also outperform their bounds.

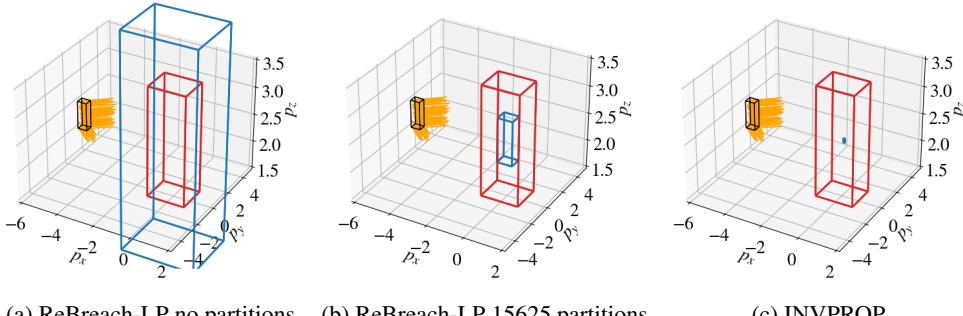

| (a) ReBreach-LP no partitions | (b) ReBreach-LP 15625 partitions | (c) INVPROP |

Figure 4: **6D Quadrotor.** We visualize the three dimensions of the 6D quadrotor that specify physical position. We consider the initial state range of the black box and simulate a few one-step trajectories, indicated by the orange lines. Our obstacle is the red box. For the three images, the blue box represents the over-approximation computed by (a) ReBreach-LP with no partitioning, (b) ReBreach-LP with 15625 partitions, and (c) INVPROP, respectively. A tighter blue box is better.

Table 2: **Quadrotor Results.** We compare the tightness of the over-approximations and run time (mean and std taken over 5 runs).

| METHOD | REBREACH-LP | REBREACH-LP | INVPROP |
|---|---|---|---|
| PARTITIONS | N/A | 15625 | N/A |
| APPROX VOL ↓ | 64 | 0.064 | 0.0000249 |
| TIME (SEC) | $11.8 \pm 2.2$ | $662.2 \pm 31.4$ | $213.8 \pm 11.1$ |

ReBreach-LP with 15625 output partitions. We run our algorithm with no branching. In Figure 4, we compare these three strategies, and we find that our over-approximation of the pre-image (in blue) is smaller than the state-of-the-art over-approximations by 257 times while being 3.29 times faster. Therefore, our algorithm scales well to higher dimensional control examples.

## 4.2 Robustness Verification

Similar to us, Yang et al. [2021b] encode the implicit output constraint of the local robustness verification query to tighten the bounds on neurons. However, they need an LP solver, which does not scale to large problems. The state-of-the-art verification toolkit $\alpha,\beta$-CROWN [Zhang et al., 2018a, Xu et al., 2021, Wang et al., 2021, Zhang et al., 2022] is able to scale to large networks, but does not currently utilize the implicit output constraint. We demonstrate the benefit of encoding the output constraint by extending $\alpha,\beta$-CROWN and comparing the performance on the benchmark used by Yang et al. [2021b] as well as benchmarks from the VNN-COMP 2023 [Brix et al., 2023].

**MNIST**  Yang et al. [2021b] provides four networks with ReLU activation functions and dense linear layers of 2/6/6/9 layers of 100/100/200/200 neurons, each trained on MNIST. For each network, 50 inputs were tested for local robustness. For the complete benchmark definition, we refer to Yang et al. [2021b]. The DeepSRGR results are taken from Yang et al. [2021b], they do not report a timeout for their experiments. Both $\alpha,\beta$-CROWN and $\alpha,\beta$-CROWN+INVPROP were run with a per-input timeout of 5 minutes. Except for the input bounds, all bounds of all layers of each network are tightened using output constraints. We report the results in Table 3. Notably, we can verify more instances than the SOTA tool $\alpha,\beta$-CROWN, in sometimes less than one fifth the average runtime. We include a more detailed comparison between the results for $\alpha,\beta$-CROWN and $\alpha,\beta$-CROWN+INVPROP in Appendix H.

**VNN-COMP '23: YOLO**  In 2023, the VNN-COMP contained YOLO as an unscored benchmark for object detection. It is a modified version of YOLOv2 [Redmon and Farhadi, 2016] with a network of 167k neurons and 5 residual layers, and is available at [Zhong, 2023, Brix, 2023]. The network processes $3 \times 52 \times 52$ images and uses convolutions, average pooling and ReLU activations. This benchmark is well suited for INVPROP, as the definition of the adversarial examples is a conjunction of constraints over the output. Therefore, a strong output constraint can be used to tighten the bounds of intermediate layers. The benchmark consists of 464 instances, of which 72 were randomly selected for the VNN-COMP. For our comparison, we remove those 348 instances that $\alpha,\beta$-CROWN verifies as robust without tightening the bounds of any intermediate layer. Those instances are verified within

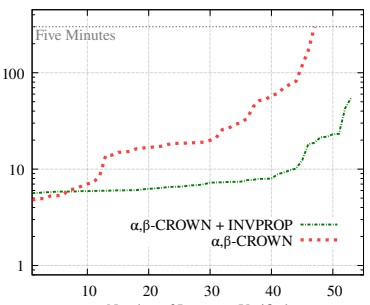

Figure 5: **YOLO Results.** Runtime comparison of $\alpha,\beta$-CROWN and $\alpha,\beta$-CROWN+INVPROP on the YOLO benchmark (167k neurons and 5 residual layers). For the comparison, only those test instances were used that could not immediately be verified by $\alpha,\beta$-CROWN without any iterative tightening of intermediate layer bounds. $\alpha,\beta$-CROWN+INVPROP can verify more properties and is faster for almost all instances.

Table 3: **MNIST Results.** Robustness verification with DeepSRGR (output constraints via LP solver), $\alpha,\beta$-CROWN (GPU support) and $\alpha,\beta$-CROWN+INVPROP (output constraints with GPU support)

| NETWORK | DEEPSRGR | | $\alpha,\beta$-CROWN | | $\alpha,\beta$-CROWN + INVPROP | |
| --- | --- | --- | --- | --- | --- | --- |
| | VERIF. | AVG. TIME [S] | VERIF. | AVG. TIME [S] | VERIF. | AVG. TIME [S] |
| FFN4 (3x100) | 35 | 781 | 45 | 7.4 | 45 | 6.4 |
| FFN5 (6x100) | 31 | 1689 | 38 | 6.7 | 39 | 8.7 |
| FFN6 (6x200) | 31 | 6178 | 29 | 5.4 | 30 | 1.0 |
| FFN7 (9x200) | 36 | 8960 | 37 | 3.2 | 40 | 1.9 |

less than four seconds each by both methods, with no room for improvement. We compare the performance of $\alpha,\beta$-CROWN and $\alpha,\beta$-CROWN + INVPROP on the remaining 116 instances in Figure 5. $\alpha,\beta$-CROWN can verify 48 instances, all other instances reach the timeout of five minutes. After extending $\alpha,\beta$-CROWN with INVPROP on the last two intermediate layers, almost all instances can be solved faster, and 6 previously timed out instances become verifiable.

### 4.3 OOD Detection

Consider the calibrated OOD detector presented as discussed in Section 2.3, encoded by a four layer MLP with 200, 200, 3, and 2 neurons. We over-approximate the set of inputs which induce a sufficiently high in-distribution (ID) confidence (measured by $\max\{y_0, y_1\} > y_2$) using 40 hyperplanes of equal slope, pictured in green in Figure 6. This set is non-convex, making the convex hull a poor over-approximation. With 4 input space branches, we get a much tighter over-approximation, as shown in the right plot. We compare the performance of our approach with and without branching over the input space with the MILP baseline (see Table 4). This demonstrates a simple proof-of-concept for how INVPROP can be used for verifying some calibration properties.

## 5 Related Work

**Formally Verified Neural Networks.** There has been a large body of work on the formal verification of neural networks, tackling the problem from the angles of Interval Bound Propagation [Gowal et al., 2018, Gehr et al., 2018], Convex Relaxations [Wong and Kolter, 2018, Shiqi et al., 2018, Salman et al., 2019, Dvijotham et al., 2018], Abstract Interpretation [Singh et al., 2018], LP [Ehlers, 2017], SDP [Raghunathan et al., 2018], SMT [Katz et al., 2017], and MILP [Tjeng et al., 2017]. However, most of this work is for forward verification (i.e., bounding the NN output given a

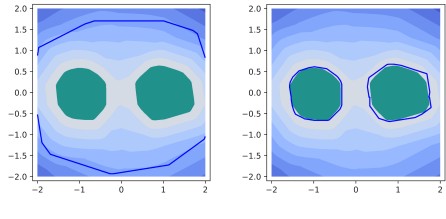

| (a) No branching | (b) Branching |
| --- | --- |

Figure 6: **OOD Detection.** Green: Empirical ID inputs (1 million samples). Blue: border of the verified preimage. Branching is union of preimages for each of 4 quadrants.

Table 4: **OOD Results.** Comparison of methods for over-approximating the preimage of the OOD region (Figure 6). A lower approximation ratio (APPROX RAT.) is better.

| METHOD | INPUT BRANCH. | APPROX RATIO | TIME (SEC) |
| --- | --- | --- | --- |
| MILP | NO | 1.47 | 1562.26 |
| INVPROP | NO | 4.39 | $3.77 \pm 0.02$ |
| INVPROP | YES | 1.14 | $12.02 \pm 0.06$ |

set of inputs). Our work strictly generalizes bound propagation as presented in Xu et al. 2021 since setting $\gamma = 0$ in Theorem 1 recovers its results.

**Formal Analysis of the Input Domain** Prior work has studied trying to determine the inverse of a neural network [Kindermann and Linden, 1990, Saad and Wunsch, 2007], though these methods are prohibitively slow. Zhang et al. [2018b] used an LP solver to compute changes to a given input that would lead to a change in the classification result while adhering to additional criteria. Yang et al. [2021b] encode the overapproximated network in an LP problem to tighten bounds by incorporating the output constraint. They require an LP solver and we compare our performance against theirs in Section 4.2. Zhong et al. [2023], Wu et al. [2022] similarly tighten intermediate bounds by using an LP solver. Dimitrov et al. [2022] compute input regions that only consist of adversarial examples (i.e., adhere to a given output constraint) that are maximized using an LP solver. Finally, a concurrent work Zhang et al. [2023] targets under- and over-approximating the preimage via LiRPA-based forward verification [Xu et al., 2020] with input-space and ReLU-space partitioning; since our work can be viewed as a generalization of LiPRA, their work may benefit from our results of tightening the intermediate layer bounds using output constraints during bound propagation.

**Formal Verification for Neural Feedback Loops.** Our control application was motivated by the growing body of work on backward reachability analysis and over-approximating states that result in a target set [Everett et al., 2021]. The original method solely utilized input constraints for deriving intermediate bounds. The later development of Everett et al. 2022 improved upon this by optimizing $l^{(0)}$ and $u^{(0)}$ with respect to output bounds but their implementation faces numerical instability when applied for many iterations. We also support the partitioning over the input space introduced by Rober et al. 2022a. The work of Vincent and Schwager 2021 explores utilizing an LP with complete neuron branching to verify control safety, which can be viewed as a domain-specific implementation of our MILP formulation of the inverse verification problem.

**Certified OOD detection.** There is a wide variety of OOD detection systems [Yang et al., 2021a, Salehi et al., 2022]. Commonly, they are evaluated empirically based on known OOD data [Wang et al., 2022, Liang et al., 2018, Chen et al., 2021]. Therefore, they fail to provide verified guarantees for their effectiveness. In fact, many OOD detection systems are susceptible to adversarial attacks [Sehwag et al., 2019, Chen et al., 2022]. Meinke et al. [2021] show how to verify robustness to adversarial examples around given input samples. Berrada et al. [2021] develop a general framework for certifying properties of output distributions of neural networks given constraints on the input distribution. However, this work is still constrained to verifying properties of the outputs given constraints (albeit probabilistic) on the inputs, and INVPROP is able to certify arbitrary regions of the input space that lead to confident predictions.

# 6 Discussion

We present the challenge of over-approximating neural network preimages and provide an efficient algorithm to solve it. By doing so, we demonstrate strong performance on multiple application areas. We believe there is a large scope for future investigation and new applications of INVPROP.

**Limitations** For higher-dimensional instances, our method would best work for box over-approximations ($2d$ hyperplanes in $d$ dimensions) and would struggle at more complicated shapes such as spheres. We expect more complex problems will benefit from a domain-specific strategy. Moreover, iteratively refining all of the intermediate bounds utilizing INVPROP faces a quadratic dependence on network depth, as opposed to a linear dependence in traditional forward verification. To mitigate this, output constraints could only be applied to layers close to the output, as we used for robustness. Finally, our branching strategy might not scale to higher dimensions, though this trade-off is well-studied [Bunel et al., 2020, Wang et al., 2021, Palma et al., 2023].

**Potential Negative Social Impact** Our work improves reliable ML through facilitating systems that provably align with practitioner expectations and we expect INVPROP to have positive societal impact. We acknowledge that our improvements may be repurposed as model attacks, though we believe the positive use cases of our technique greatly outweigh current speculation of misusage.

**Acknowledgements** We thank Michael Everett and Nicholas Rober for helpful discussion and feedback on the paper. Huan Zhang acknowledges the support of the Schmidt Futures AI2050 Early Career Fellowship.

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

## A  OOD Detection

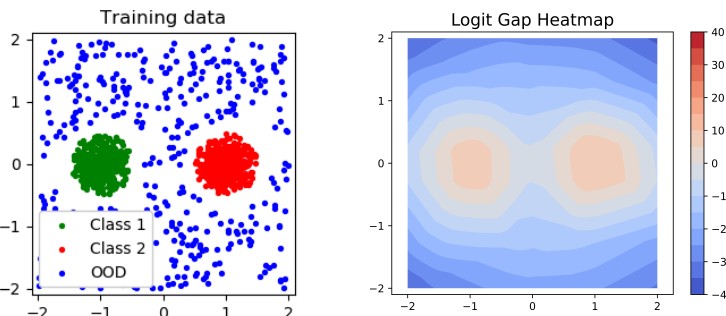

Figure 7: To improve our model's ability to detect data outside the training distribution (of green and red points), we randomly sample points that are far from every training point. In the right contour, we see that the model's confidence of in-distribution increases as we approach the centers of the distribution, demonstrating our model is more calibrated.

For the OOD benchmark, we train a neural network on two clusters of data. Standard training would lead to an uncalibrated predictor, we add in OOD data through sampling the input space and rejecting inputs that are too close to any training data point. Training a classifier on this dataset induces the contour on the right, which is a calibrated classifier we can certify properties over. For our experiments, we consider a high-confidence output as 90% confidence of the data being in-distribution.

## B  Example

**Example 1.** Consider the toy neural network in Figure 8 under $\mathcal{X} = [-2, 2]$ and $\mathcal{S}_{\text{out}} = \{y : 1 \leq y \leq 1.02\}$. Suppose we wanted to find the tightest $[l_0^{(1)}, u_0^{(1)}]$ (bounds on $x_0^{(1)}$). If we only enforce the constraints preceding the ReLU (standard bound propagation), we get $[-2, 2]$. If we only enforce the constraints following the ReLU, we get $(-\infty, 1.02]$. However, when we utilize all of the constraints, we find that the true intermediate bounds are $[0, 0.01]$, which is over 100 times tighter than the intersection of the two previous methods (all derivations provided below). Therefore, new techniques are necessary for optimizing intermediate bounds in this setting.

### B.1  Deriving Intermediate Bounds for Example B

**Preceding Constraints.**   The linear layer gives us $x_0^{(1)} = \hat{x}_0^{(0)}$. Since the only constraint we have is $\hat{x}_0^{(0)} \in [-2, 2]$, the tightest intermediate bounds we can derive are $[-2, 2]$.

**Following Constraints.**   Since the output of a ReLU is non-negative, we know that $\hat{x}_0^{(1)}, \hat{x}_1^{(1)} \geq 0$. Since $x_0^{(2)} = \hat{x}_0^{(1)} + \hat{x}_1^{(1)}$ and the output constraint enforces $x_0^{(2)} \leq 1.02$, we derive that $\hat{x}_0^{(1)} \leq 1.02$. If we set $\hat{x}_0^{(1)} = 1.02 - \hat{x}_1^{(1)}$, we see that $\hat{x}_0^{(1)}$ can take the entire interval $[0, 1.02]$. Since the only constraint we have on $x_0^{(1)}$ is that $\mathsf{ReLU}(\hat{x}_0^{(1)}) = x^{(1)}$, our desired interval is the preimage of $[0, 1.02]$. This means that $x_0^{(1)}$ can take any value in the interval $(-\infty, 1.02]$.

**All Constraints.**   We first note that by the first linear layer, we have $x_1^{(1)} = x_0^{(1)} + 1$. Therefore, if $x_0^{(1)}$ is less than 0, then $x_1^{(1)}$ is less than 1, which means $\mathsf{ReLU}(x_0^{(1)}) + \mathsf{ReLU}(x_1^{(1)})$ is less than 1, which contradicts the output constraint. If $x_0^{(1)}$ is always non-negative, then we have that the output is equivalent to $\mathsf{ReLU}(x_0^{(1)}) + \mathsf{ReLU}(x_0^{(1)} + 1) = 2x_0^{(1)} + 1$. Therefore, the output constraint implies $x_0^{(1)} \leq 0.01$. Any $x_0^{(1)} \in [0, 0.01]$ is achievable by setting the input to the desired value.

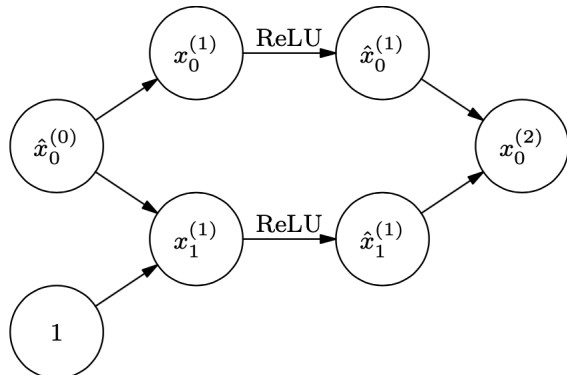

Figure 8: A simple neural network from $\mathbb{R}$ to $\mathbb{R}$. Every node is the sum of its incoming nodes unless the edge is labeled ReLU.

## C Optimizing Intermediate Bounds

We present a generalized theorem which provides a lower bound for any linear combination of $\hat{\boldsymbol{x}}^{(0)}$ and $\boldsymbol{x}^{(i)}$. We prove this theorem in Appendix D. We note that by selecting the coefficients of these variables (named $\boldsymbol{c}^{(i)}$ for $i \in \{0, 1, \ldots, L-1\}$), we recover the two following functionalities which we name $g_{\boldsymbol{c}}(\boldsymbol{\alpha}, \boldsymbol{\gamma})$ and $g_{i,j}^b(\boldsymbol{\alpha}, \boldsymbol{\gamma})$ (respectively).

- When the only nonzero coefficients are $\boldsymbol{c}^{(0)} = \boldsymbol{c}$, we recover Theorem 1. We refer to this bound as $g_{\boldsymbol{c}}(\boldsymbol{\alpha}, \boldsymbol{\gamma})$

- When the only nonzero coefficient is $c_j^{(i)} = 1$, we can lower bound $l_j^{(i)}$ by $g_{i,j}^{\text{lower}}(\boldsymbol{\alpha}, \boldsymbol{\gamma})$. If we set $c_j^{(i)} = -1$, we can upper bound $u_j^{(i)}$ by $-g_{i,j}^{\text{upper}}(\boldsymbol{\alpha}, \boldsymbol{\gamma})$.

With this, we present our generalized theorem.

**Theorem 2** (Lower-bounding combination of neurons). *Given an output set $\mathcal{S}_{out} = \{\boldsymbol{y} : \mathbf{H}\boldsymbol{y}+\mathbf{d} \leq \mathbf{0}\}$ and vector $\boldsymbol{c}$, $g(\boldsymbol{\alpha}, \boldsymbol{\gamma})$ is a lower bound to the linear program in (3) with objective $\boldsymbol{c}^{(0)\top}\hat{\boldsymbol{x}}^{(0)} + \sum_{i=1}^{L-1} \boldsymbol{c}^{(i)\top}\boldsymbol{x}^{(i)}$ for $\mathbf{0} \leq \boldsymbol{\alpha} \leq \mathbf{1}$, $\boldsymbol{\gamma} \geq \mathbf{0}$, and $g$ defined via*

$$g(\boldsymbol{\alpha}, \boldsymbol{\gamma}) = \left[\boldsymbol{c}^{(0)\top} - \boldsymbol{\nu}^{(1)\top}\mathbf{W}^{(1)}\right]_+ \boldsymbol{l}^{(0)} - \left[\boldsymbol{c}^{(0)\top} - \boldsymbol{\nu}^{(1)\top}\mathbf{W}^{(1)}\right]_- \boldsymbol{u}^{(0)}$$
$$- \sum_{i=1}^{L} \boldsymbol{\nu}^{(i)\top}\mathbf{b}^{(i)} + \sum_{i=1}^{L-1} \sum_{j \in \mathcal{I}^{\pm(i)}} \left[\frac{u_j^{(i)} l_j^{(i)} [\hat{\nu}_j^{(i)}]_+}{u_i^{(j)} - l_i^{(j)}}\right]$$

*where every term can be directly recursively computed via*

$$\mathcal{I}^{-(i)} = \{j : u_j^{(i)} \leq 0\}$$
$$\mathcal{I}^{+(i)} = \{j : l_j^{(i)} \geq 0\}$$
$$\mathcal{I}^{\pm(i)} = \{j : l_j^{(i)} < 0 < u_j^{(i)}\}$$
$$\boldsymbol{\nu}^{(L)} = -\boldsymbol{\gamma}$$
$$\hat{\nu}_j^{(i)} = \boldsymbol{\nu}^{(i+1)\top}\mathbf{W}_{:,j}^{(i+1)}$$
$$\nu_j^{(i)} = \begin{cases} \hat{\nu}_j^{(i)} - c_j^{(i)}, & j \in \mathcal{I}^{+(i)} \\ -c_j^{(i)}, & j \in \mathcal{I}^{-(i)} \\ \frac{u_j^{(i)}}{u_i^{(j)} - l_i^{(j)}}[\hat{\nu}_j^{(i)}]_+ - \alpha_j^{(i)}[\hat{\nu}_j^{(i)}]_- - c_j^{(i)}, & j \in \mathcal{I}^{\pm(i)} \end{cases}$$

# D    Dual Derivation

We first incorporate the output constraint $\mathbf{H}x^{(L)} + \mathbf{d} \leq \mathbf{0}$ by folding in this linear transformation of the output into the final linear layer of the network, as customary in prior work [Wang et al., 2021, Zhang et al., 2022].

We now prove the generalized theorem as described in C. As a conceptual overview of our proof, we take the Lagrange dual of the linear program to derive an unconstrained optimization problem. From here, we derive constraints that must be satisfied in the $\max \min$ formulation. These constraints yield the bound propagation procedure we display in Theorem 1.

We start with the convex relaxation.

$$
\begin{aligned}
\min_{\boldsymbol{x}, \hat{\boldsymbol{x}}} \quad & \boldsymbol{c}^{(0)\top} \hat{\boldsymbol{x}}^{(0)} + \sum_{i=1}^{L-1} \boldsymbol{c}^{(i)\top} \boldsymbol{x}^{(i)} \\
\text{s.t.} \quad & \boldsymbol{l}^{(0)} \leq \hat{\boldsymbol{x}}^{(0)} \leq \boldsymbol{u}^{(0)} \\
& \boldsymbol{x}^{(L)} \leq \mathbf{0}; \\
& \boldsymbol{x}^{(i)} = \mathbf{W}^{(i)} \hat{\boldsymbol{x}}^{(i-1)} + \mathbf{b}^{(i)}; \quad i \in [L], \\
& \hat{x}_j^{(i)} \geq 0; j \in \mathcal{I}^{\pm(i)} \\
& \hat{x}_j^{(i)} \geq x_j^{(i)}; j \in \mathcal{I}^{\pm(i)} \\
& (u_j^{(i)} - l_j^{(i)}) \hat{x}_j^{(i)} \leq u_j^{(i)} x_j^{(i)} - u_j^{(i)} l_j^{(i)}; j \in \mathcal{I}^{\pm(i)} \\
& \hat{x}_j^{(i)} = x_j^{(i)}; j \in \mathcal{I}^{+(i)} \\
& \hat{x}_j^{(i)} = 0; j \in \mathcal{I}^{-(i)}
\end{aligned}
$$

From this, we take the Lagrange dual of most of the constraints. In doing so, we introduce a new dual variable for each constraint. Note that we do not dualize every constraint since they can easily be dealt with later.

$$
\begin{aligned}
\min_{\boldsymbol{x}, \hat{\boldsymbol{x}}} \max_{\boldsymbol{\nu}, \boldsymbol{\mu}, \boldsymbol{\tau}, \boldsymbol{\gamma}, \boldsymbol{\lambda}} \quad & \boldsymbol{c}^{(0)\top} \hat{\boldsymbol{x}}^{(0)} + \sum_{i=1}^{L-1} \boldsymbol{c}^{(i)\top} \boldsymbol{x}^{(i)} + \boldsymbol{\gamma}^\top \boldsymbol{x}^{(L)} + \sum_{i=1}^{L} \boldsymbol{\nu}^{(i)\top} \left( \boldsymbol{x}^{(i)} - \mathbf{W}^{(i)} \hat{\boldsymbol{x}}^{(i-1)} - \mathbf{b}^{(i)} \right) \\
& + \sum_{i=1}^{L-1} \sum_{j \in \mathcal{I}^{\pm(i)}} \left[ \mu_j^{(i)} \left( -\hat{x}_j^{(i)} \right) + \tau_j^{(i)} \left( x_j^{(i)} - \hat{x}_j^{(i)} \right) + \lambda_j^{(i)} \left( (u_j^{(i)} - l_j^{(i)}) \hat{x}_j^{(i)} - u_j^{(i)} x_j^{(i)} + u_j^{(i)} l_j^{(i)} \right) \right] \\
\text{s.t. } & \boldsymbol{l}^{(0)} \leq \hat{\boldsymbol{x}}^{(0)} \leq \boldsymbol{u}^{(0)}; \quad \hat{x}_j^{(i)} = 0, j \in \mathcal{I}^{-(i)}; \quad \hat{x}_j^{(i)} = x_j^{(i)}, j \in \mathcal{I}^{+(i)} \\
& \boldsymbol{\mu} \geq 0; \quad \boldsymbol{\tau} \geq 0; \quad \boldsymbol{\gamma} \geq 0; \quad \boldsymbol{\lambda} \geq 0
\end{aligned}
$$

Since we took the dual of a linear program, the solution to the min-max optimization is equivalent to the solution of the max-min optimization by strong duality. Therefore, we can solve the following program (equivalent up to rearrangement).

$$\max_{\boldsymbol{\nu},\boldsymbol{\mu},\boldsymbol{\tau},\boldsymbol{\gamma},\boldsymbol{\lambda}} \min_{\boldsymbol{x},\hat{\boldsymbol{x}}} \quad \left(\boldsymbol{\nu}^{(L)}+\boldsymbol{\gamma}\right)^{\top} \boldsymbol{x}^{(L)} + \left(\boldsymbol{c}^{(0)\top} - \boldsymbol{\nu}^{(1)\top}\mathbf{W}^{(1)}\right)\hat{\boldsymbol{x}}^{(0)}$$

$$+ \sum_{i=1}^{L-1}\sum_{j\in\mathcal{I}^{+(i)}} \left(\nu_j^{(i)} - \boldsymbol{\nu}^{(i+1)\top}\mathbf{W}_{:,j}^{(i+1)} + c_j^{(i)}\right) x_j^{(i)} + \sum_{i=1}^{L-1}\sum_{j\in\mathcal{I}^{-(i)}} \left(\nu_j^{(i)} + c_j^{(i)}\right) x_j^{(i)}$$

$$+ \sum_{i=1}^{L-1}\sum_{j\in\mathcal{I}^{\pm(i)}} \left[\left(\nu_j^{(i)} + \tau_j^{(i)} - \lambda_j^{(i)}u_j^{(i)} + c_j^{(i)}\right) x_j^{(i)}\right.$$

$$\left. + \left(-\boldsymbol{\nu}^{(i+1)\top}\mathbf{W}_{:,j}^{(i+1)} - \mu_j^{(i)} - \tau_j^{(i)} + (u_j^{(i)} - l_j^{(i)})\lambda_j^{(i)}\right)\hat{x}_j^{(i)}\right]$$

$$- \sum_{i=1}^{L}\boldsymbol{\nu}^{(i)\top}\mathbf{b}^{(i)} + \sum_{i=1}^{L-1}\sum_{j\in\mathcal{I}^{\pm(i)}} \lambda_j^{(i)}u_j^{(i)}l_j^{(i)}$$

$$\text{s.t. } \boldsymbol{l}^{(0)} \le \hat{\boldsymbol{x}}^{(0)} \le \boldsymbol{u}^{(0)}$$

$$\boldsymbol{\mu} \ge 0; \quad \boldsymbol{\tau} \ge 0; \quad \boldsymbol{\gamma} \ge 0; \quad \boldsymbol{\lambda} \ge 0$$

To minimize $\left(\boldsymbol{c}^{(0)\top} - \boldsymbol{\nu}^{(1)\top}\mathbf{W}^{(1)}\right)\hat{\boldsymbol{x}}^{(0)}$ subject to $\boldsymbol{l}^{(0)} \le \hat{\boldsymbol{x}}^{(0)} \le \boldsymbol{u}^{(0)}$, we can consider the choice we must make in each dimension. If the $j$th entry of the coefficient is positive, we should set $\hat{x}_j^{(0)} = l_j^{(0)}$. Otherwise, we should set $\hat{x}_j^{(0)} = u_j^{(0)}$.

$$\max_{\boldsymbol{\nu},\boldsymbol{\mu},\boldsymbol{\tau},\boldsymbol{\gamma},\boldsymbol{\lambda}} \min_{\boldsymbol{x},\hat{\boldsymbol{x}}} \quad \left(\boldsymbol{\nu}^{(L)}+\boldsymbol{\gamma}\right)^{\top} \boldsymbol{x}^{(L)} + \left[\boldsymbol{c}^{(0)\top} - \boldsymbol{\nu}^{(1)\top}\mathbf{W}^{(1)}\right]_{+}\boldsymbol{l}^{(0)} - \left[\boldsymbol{c}^{(0)\top} - \boldsymbol{\nu}^{(1)\top}\mathbf{W}^{(1)}\right]_{-}\boldsymbol{u}^{(0)}$$

$$+ \sum_{i=1}^{L-1}\sum_{j\in\mathcal{I}^{+(i)}} \left(\nu_j^{(i)} - \boldsymbol{\nu}^{(i+1)\top}\mathbf{W}_{:,j}^{(i+1)} + c_j^{(i)}\right) x_j^{(i)} + \sum_{i=1}^{L-1}\sum_{j\in\mathcal{I}^{-(i)}} \left(\nu_j^{(i)} + c_j^{(i)}\right) x_j^{(i)}$$

$$+ \sum_{i=1}^{L-1}\sum_{j\in\mathcal{I}^{\pm(i)}} \left[\left(\nu_j^{(i)} + \tau_j^{(i)} - \lambda_j^{(i)}u_j^{(i)} + c_j^{(i)}\right) x_j^{(i)}\right.$$

$$\left. + \left(-\boldsymbol{\nu}^{(i+1)\top}\mathbf{W}_{:,j}^{(i+1)} - \mu_j^{(i)} - \tau_j^{(i)} + (u_j^{(i)} - l_j^{(i)})\lambda_j^{(i)}\right)\hat{x}_j^{(i)}\right]$$

$$- \sum_{i=1}^{L}\boldsymbol{\nu}^{(i)\top}\mathbf{b}^{(i)} + \sum_{i=1}^{L-1}\sum_{j\in\mathcal{I}^{\pm(i)}} \lambda_j^{(i)}u_j^{(i)}l_j^{(i)}$$

$$\text{s.t. } \boldsymbol{\mu} \ge 0; \quad \boldsymbol{\tau} \ge 0; \quad \boldsymbol{\gamma} \ge 0; \quad \boldsymbol{\lambda} \ge 0$$

From here, we note that the variables $\boldsymbol{x}$ or $\hat{\boldsymbol{x}}$ are unconstrained variables. Therefore, if any of their coefficients are nonzero, the inner minimization can immediately drive its value to $-\infty$. As such, the outer maximization must set all of these coefficients to zero. Therefore, we can derive constraints from this restructured optimization and remove the free variables $\boldsymbol{x}, \hat{\boldsymbol{x}}$.

$$\max_{\boldsymbol{\nu},\boldsymbol{\mu},\boldsymbol{\tau},\boldsymbol{\gamma},\boldsymbol{\lambda}} \quad \left[\boldsymbol{c}^{(0)\top} - \boldsymbol{\nu}^{(1)\top}\mathbf{W}^{(1)}\right]_+ \boldsymbol{l}^{(0)} - \left[\boldsymbol{c}^{(0)\top} - \boldsymbol{\nu}^{(1)\top}\mathbf{W}^{(1)}\right]_- \boldsymbol{u}^{(0)} - \sum_{i=1}^{L} \boldsymbol{\nu}^{(i)\top}\mathbf{b}^{(i)}$$

$$+ \sum_{i=1}^{L-1} \sum_{j\in\mathcal{I}^{\pm(i)}} \lambda_j^{(i)} u_j^{(i)} l_j^{(i)}$$

$$\text{s.t.} \quad \boldsymbol{\nu}^{(L)} = -\boldsymbol{\gamma}$$

$$\nu_j^{(i)} = \boldsymbol{\nu}^{(i+1)\top}\mathbf{W}_{:,j}^{(i+1)} - c_j^{(i)}, j \in \mathcal{I}^{+(i)}$$

$$\nu_j^{(i)} = -c_j^{(i)}, j \in \mathcal{I}^{-(i)}$$

$$\nu_j^{(i)} = \lambda_j^{(i)} u_j^{(i)} - \tau_j^{(i)} - c_j^{(i)}, j \in \mathcal{I}^{\pm(i)}$$

$$\boldsymbol{\nu}^{(i+1)\top}\mathbf{W}_{:,j}^{(i+1)} = (u_j^{(i)} - l_j^{(i)})\lambda_j^{(i)} - \left(\mu_j^{(i)} + \tau_j^{(i)}\right), j \in \mathcal{I}^{\pm(i)}$$

$$\boldsymbol{\mu} \geq 0; \quad \boldsymbol{\tau} \geq 0; \quad \boldsymbol{\gamma} \geq 0; \quad \boldsymbol{\lambda} \geq 0$$

For the following, we define $\hat{\nu}_j^{(i)} = \boldsymbol{\nu}^{(i+1)\top}\mathbf{W}_{:,j}^{(i+1)}$. We note that since the upper and lower bounds of the neuron relaxation can not be tight simulataneously, at least one of $(u_j^{(i)} - l_j^{(i)})\lambda_j^{(i)}$ and $\mu_j^{(i)} + \tau_j^{(i)}$ must be non-zero. Therefore, we can write them as $(u_j^{(i)} - l_j^{(i)})\lambda_j^{(i)} = [\hat{\nu}_j^{(i)}]_+$ and $\mu_j^{(i)} + \tau_j^{(i)} = [\hat{\nu}_j^{(i)}]_-$. We can then use the fact that $\tau_j^{(i)}$ lies in the interval 0 and $\hat{\nu}_j^{(i)}$ to get the following bound propagation procedure.

$$\max_{\boldsymbol{\nu},\boldsymbol{\alpha},\boldsymbol{\gamma}} \quad \left[\boldsymbol{c}^{(0)\top} - \boldsymbol{\nu}^{(1)\top}\mathbf{W}^{(1)}\right]_+ \boldsymbol{l}^{(0)} - \left[\boldsymbol{c}^{(0)\top} - \boldsymbol{\nu}^{(1)\top}\mathbf{W}^{(1)}\right]_- \boldsymbol{u}^{(0)} - \sum_{i=1}^{L} \boldsymbol{\nu}^{(i)\top}\mathbf{b}^{(i)}$$

$$+ \sum_{i=1}^{L-1} \sum_{j\in\mathcal{I}^{\pm(i)}} \left[ \frac{u_j^{(i)} l_j^{(i)} [\hat{\nu}_j^{(i)}]_+}{u_i^{(j)} - l_i^{(j)}} \right]$$

$$\text{s.t.} \quad \boldsymbol{\nu}^{(L)} = -\boldsymbol{\gamma}$$

$$\nu_j^{(i)} = \boldsymbol{\nu}^{(i+1)\top}\mathbf{W}_{:,j}^{(i+1)} - c_j^{(i)}, j \in \mathcal{I}^{+(i)}$$

$$\nu_j^{(i)} = -c_j^{(i)}, j \in \mathcal{I}^{-(i)}$$

$$\hat{\nu}_j^{(i)} = \boldsymbol{\nu}^{(i+1)\top}\mathbf{W}_{:,j}^{(i+1)}$$

$$\nu_j^{(i)} = \frac{u_j^{(i)}}{u_i^{(j)} - l_i^{(j)}}[\hat{\nu}_j^{(i)}]_+ - \alpha_j^{(i)}[\hat{\nu}_j^{(i)}]_- - c_j^{(i)}, j \in \mathcal{I}^{\pm(i)}$$

$$\alpha_j^{(i)} \in [0,1]; \quad \boldsymbol{\gamma} \geq 0$$

In this program, $\alpha_j^{(i)}$ are optimizable parameters controlling the relaxation of neuron $j$ in layer $i$, similar to the ones appearing in Xu et al. 2021. $\boldsymbol{\gamma}$, as discussed in the body of the paper, is the parameter which enforces the output constraint throughout this entire bound propagation procedure.

## E Connection to Forward Verification for Intermediate Bounds

We consider the general objective presented in C and aim to minimize $\boldsymbol{c}^{(0)\top}\hat{\boldsymbol{x}}^{(0)} + \sum_{i=1}^{L-1} \boldsymbol{c}^{(i)\top}\boldsymbol{x}^{(i)}$

$$\min_{\boldsymbol{x}} \quad \boldsymbol{c}^{(0)\top}\hat{\boldsymbol{x}}^{(0)} + \sum_{i=1}^{L-1} \boldsymbol{c}^{(i)\top}\boldsymbol{x}^{(i)}$$

$$\text{s.t.} \quad \boldsymbol{x} \in \mathcal{X}; \quad \mathbf{H}f(\boldsymbol{x}) + \mathbf{d} \leq 0$$

If we take the dual of the constraint, then we get the program

$$\min_{\boldsymbol{x}} \quad \boldsymbol{c}^{(0)\top}\hat{\boldsymbol{x}}^{(0)} + \sum_{i=1}^{L-1} \boldsymbol{c}^{(i)\top}\boldsymbol{x}^{(i)} + \boldsymbol{\gamma}\left(\mathbf{H}f(\boldsymbol{x}) + \mathbf{d}\right)$$
$$\text{s.t.} \quad \boldsymbol{x} \in \mathcal{X}; \quad \boldsymbol{\gamma} \geq 0$$

We note that the objective here can be expressed as a neural network with residual stream $f(\boldsymbol{x})$ and a skip connection with linear weight $\boldsymbol{c}^{(i)}$ from the pre-activations of layer $i$ to the final output for every layer. In theory, we could construct this neural network and directly pass it to a forward verification tool which could iteratively tighten all bounds for a solution.

# F   Implementation

As stated in Algorithm 1, INVPROP first initializes bounds for all layers using some computationally cheap technique. Based on the input bounds given by $\mathcal{X}$, we first compute intermediate bounds using interval propagation and then tighten them based on the reverse symbolic interval propagation (RSIP) technique Singh et al. [2019]. While INVPROP will iteratively tighten those bounds over time, we found RSIP to reduce the total necessary runtime significantly compared to an initialization based solely on interval propagation. We initialize all $\boldsymbol{\alpha}$ with 0.5 and all $\boldsymbol{\gamma}$ with 0.025.

The optimization is performed for all lower and upper bounds of all neurons in each layer in parallel, starting with the last layer and moving forward to the input layer. The bounds of the cutting hyperplanes are optimized last, together with the bounds on the input neurons. The improvements on $\boldsymbol{c}^\top\boldsymbol{x}$ are measured every 10th iteration. Before doing so, the bounds of the hyperplanes are tightened for 10 extra steps to improve their precision. We detect convergence by monitoring $\boldsymbol{c}^\top\boldsymbol{x}$. If successive iterations see minimal improvement, we stop or branch on the input space. All cutting hyperplanes are evenly distributed to maximize their information gain. For 40 hyperplanes in a 2D input space, we rotate each plane by 9°. All reported runtimes under 10 minutes are computed as the average of five runs.

## F.1   Encoding Non-Linear Constraints

To support non-linear output sets, such as the maximum operation used in the OOD example in Section 4.3, the non-linearity needs to be encoded, such that it can be expressed as linear constraints over the modified network. We rewrite $\max(y_1, y_2) = \max(y_1 - y_2, 0) + y_2$ and add an additional ReLU layer for this operation. Note that to pass $y_2$ through this layer without modifying it, one can either write $y_2 = \max(y_2, 0) - \max(-y_2, 0)$, or $y_2 = \max(y_2 - M, 0) + M$ where $M$ is the lower bound of all possible $y_2$. We find that avoiding the additional ReLU relaxations that would occur for the first approach is beneficial for the optimization and compute a lower bound of $M$ using interval propagation. As $y_3$ (for the OOD class) should not be changed by this operation either, we apply the same trick of subtracting and adding its lower bound.

## F.2   Control Benchmark Encoding

We encode the entire control formula $\boldsymbol{x} = \mathbf{A}\boldsymbol{x} + \mathbf{B}\boldsymbol{u}$ as one feedforward network by encoding the residual connection as regular fully connected layers. To this end, we use the same technique described in Section F.1 to shift the bounds into the positive regime, feed them forward and then shift them back.

To compute $\mathcal{S}_{\text{over}}$ for a timestep $t > 1$, we first stack the network $t$ times, then simplify it by merging consecutive linear layers. All bounds of layers not affected by this merging that also appear in the network for $t-1$ timesteps are reused. Their bounds are already tight enough and are not optimized further. Note that this is different than using the previous $\mathcal{S}_{\text{over}}$ as the new target region and using an unstacked network: All bounds of the new layers are still optimized w.r.t. the precise target $\mathcal{S}_{\text{out}}$. Therefore, we do not suffer from accumulating inaccuracies.

# G   Hardware

For the control benchmark, all experiments were performed on a Dual Xeon Gold 6138.

For the OOD benchmark, all experiments were performed on an Intel Xeon Platinum 8160 processor using 8 cores and 40GB RAM, as well as a V100-SXM2 GPU.

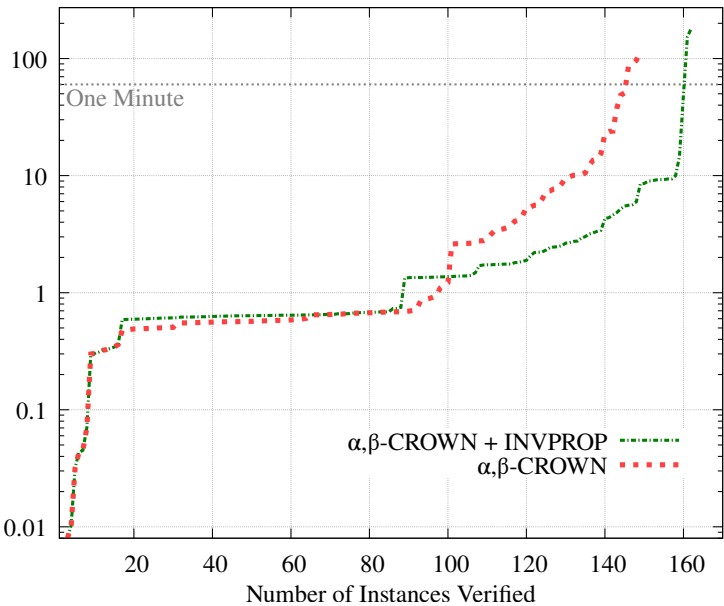

Figure 9: Detailed runtime comparison of $\alpha,\beta$-CROWN and $\alpha,\beta$-CROWN + INVPROP. Except for a few instances, $\alpha,\beta$-CROWN extended with INVPROP can verify the properties faster, and can prove robustness for some instances that cause a timeout for pure $\alpha,\beta$-CROWN. The timeout per instance is 5 minutes.

For the MNIST and YOLO robostness verification benchmarks, an AWS instance of type g5.2xlarge was used, with is equipped with a AMD EPYC 7R32 processor with 8 cores, 32GB RAM, as well as an A10G GPU.

## H    Robustness Verification

For the comparison in Section 4.2, we have implemented the concept of INVPROP in $\alpha,\beta$-CROWN. Figure 9 compares the performance of $\alpha,\beta$-CROWN and $\alpha,\beta$-CROWN + INVPROP on the MNIST benchmark defined by Yang et al. [2021b].

