# OpenReview forum: "Provably Bounding Neural Network Preimages"
_NeurIPS.cc/2023/Conference — NeurIPS 2023 spotlight_

### Official Review · Reviewer_XnFH · 2023-06-08

**Soundness:** 3 good
**Presentation:** 4 excellent
**Contribution:** 2 fair
**Rating:** 7
**Confidence:** 3

**Summary:**

This paper presents a novel approach to bounding the preimages of neural networks using a sequence of relaxations. Results are provided for a toy controls problem and a robustness verification task.

**Strengths:**

1. The paper addresses an important recent developement in the literature.
2. The paper is well presented and motivated. I like the direct correspondence between the visualization in Figure 1 and the derivation of the method in Section 3.
3. The proposed approach is novel and has some promising results for toy problems. I believe all the steps in the derivation of the approach are correct, although I have not checked them carefully.

**Weaknesses:**

1. My main concern is that the main experiment that the approach is evaluated on (reachability analysis) is very small (12 + 7 + 2 neurons), even by verification standards. It is true that these optimization-problem flavored verification approaches are generally evaluated on small networks, ab-CROWN for example can be evaluated for substantially larger networks on datasets like CIFAR-10. Given that these approaches incur exponential runtime with network size, I am skeptical that they can ever be relevant for networks of a practical size.
2. Similarly, many proposed details scale poorly with input data dimension (as mentioned in the limitations). I believe that the number of sampled c would need to grow exponentially with input dimension, as would the branch-and-bound scheme.
3. I recommend the authors tone down their presentation somewhat. In 2.2, it should be claimed that only an approximation of the convex hull is computed (not the exact convex hull). Also referring to the controls problem as an "established benchmark" is somewhat of a stretch as this is a recent unpublished preprint with only a couple citations.

**Questions:**

1. I am not sure why the controls benchmark task is referred to as a "robot moving on the floor of a room." This is a double integrator task, as described in the cited work that originally presented the problem. Of course this can be considered a planar system with strange dynamics, but it's not clear why that would be done when the dynamics have a classic double integrator structure.
2. I'm confused on the discrepancy between the reported results for the baseline method. Specifically, the approximation ratio of ReBreach is about 4000, which is orders of magnitude higher than what I would expect based on the results in the original paper (errors less than 10x the true volume). What is causing this discrepancy?
3. In Figure 3, on the left side without input splitting, why is the convex hull so loose? I would expect it to tightly wrap around the green circles.

**Limitations:**

The main limitations are mentioned explicitly.

---

> ### Author Rebuttal · Authors · 2023-08-10
>
> Thank you for the feedback!  Thanks to your review, we have demonstrated our scalability through much more difficult robustness verification (including **8112-dimensional inputs** and **167k neurons**) and tested another reachability analysis problem of a **6-dimensional state quadrotor**.
>
> > Given that these approaches incur exponential runtime with network size, I am skeptical that they can ever be relevant for networks of a practical size.
>
> In light of these suggestions, we have run new experiments to highlight the strength of our method. In the verified robustness setting, our method handles **8112-dimensional inputs** for **167k neuron** networks. In the control setting, our method handles a **6-dimensional quadrotor control** problem. Furthermore, we provide more details on scalability across dimensions/neurons/branching in our new limitations section. Please see our new results and limitations in our general response and PDF.
>
> > Similarly, many proposed details scale poorly with input data dimension (as mentioned in the limitations). I believe that the number of sampled c would need to grow exponentially with input dimension, as would the branch-and-bound scheme.
>
> The selection of hyperplanes is a domain-specific consideration and is a separate consideration from the INVPROP algorithm. For example, in higher dimensional applications, using two planes per input dimension, the positive and negative unit vectors, gives a box over-approximation of the pre-image. We find that this scales well in practice for robustness verification tasks (as it offers improvements for robustness verification in dimension 8112 and control verification in dimension 6) and scales well in theory since the number of hyperplanes necessary would be linear in the input dimension.
>
> In our new results, we integrate INVPROP with alpha-beta-CROWN and show that it works well in the case of robustness verification. Utilizing Branch and Bound for verification is independent of this work, and the trade-off between the tightness of verification and computational constraints is well-studied in prior literature.
>
> We now provide this discussion in our limitations section.
>
> >I recommend the authors tone down their presentation somewhat. In 2.2, it should be claimed that only an approximation of the convex hull is computed (not the exact convex hull). Also referring to the controls problem as an "established benchmark" is somewhat of a stretch as this is a recent unpublished preprint with only a couple citations.
>
> Thank you for this suggestion. We have toned down the claim to "we aim to over-approximation the convex hull". As for the benchmark, it has been covered in a body of other papers outside of the DRIP preprint (the benchmark is from "nfl_veripy: Formal Verification of Neural Feedback Loops (NFLs)", but we cannot leave a link here per NeurIPS rule).
>
> >I am not sure why the controls benchmark task is referred to as a "robot moving on the floor of a room." This is a double integrator task, as described in the cited work that originally presented the problem. Of course this can be considered a planar system with strange dynamics, but it's not clear why that would be done when the dynamics have a classic double integrator structure.
>
> We agree that the robot scenario does not properly highlight our story. We change this story to the DoubleIntegrator task we target later in the paper (since these dynamics are precisely the ones we verify in Section 4.1).
>
> >I'm confused on the discrepancy between the reported results for the baseline method. Specifically, the approximation ratio of ReBreach is about 4000, which is orders of magnitude higher than what I would expect based on the results in the original paper (errors less than 10x the true volume). What is causing this discrepancy?
>
> The original paper only reports results for five timesteps back. We find this benchmark too simple, so we repeat this benchmark for ten timesteps back, where exponential error accrues due to looseness in the over-approximation.
>
> >In Figure 3, on the left side without input splitting, why is the convex hull so loose? I would expect it to tightly wrap around the green circles.
>
> One possibility is that the convex over-approximation needs neuron branching or further input/output branching to get the tightest hyperplanes. Another possibility is that we did not run our "without splitting" algorithm for sufficient iterations, so it did not converge to the true convex hull.
>
> –
>
> We hope that we’ve addressed the proposed concerns and would be more than happy to answer additional questions!

---

> > ### Comment · Reviewer_XnFH · 2023-08-10
> >
> > Thanks for the clarification. I'm a little confused on the quadrotor example. What exactly are you computing an over approximation of? The quadrotor starts in the black box, and the orange trajectories leave the black box a little bit, but then the blue box is far away from the trajectories, and for invprop appears to lie inside the obstacles? I'm not really sure what I'm looking at here.
> >
> > Also, is this a quadrotor or a linearized quadrotor plant?

---

> > > ### Author Response · Authors · 2023-08-12
> > >
> > > Your analysis is correct. The black box and orange lines are a simple demonstration of the quadrotor movement; note that these are not supplied to any of the algorithms. The red box denotes an obstacle (which we don't want to collide into at time = 1) and the blue box represents an over-approximation of the states which result in entering this red box (over states at time = 0). Since the blue and the black box do not overlap (both representing possibilities at time = 0), it means that we have certified that starting from the black box will never enter the red box. Since our over-approximation forms a subset of the obstacle, we simultaneously prove the stronger claim that one will not collide with the obstacle for any starting point outside of it, which can not be proven by ReBreach-LP with no partitions. We note that these choices are made by prior work of [1] (Section V, Subsection D) and the code to reproduce their experiments, and we keep them so we can provide a fair comparison to prior work.
> > >
> > > As for more details about the quadrotor, it is a 6-dimensional linearized quadrotor model where each coordinate is represented by a double integrator. The precise experimental settings such as the dynamics can be found in [1](Section V, Subsection D).
> > >
> > > Please feel free to let us know if it is still unclear or if you have any additional questions. Thank you!

---

> > > > ### Comment · Reviewer_XnFH · 2023-08-14
> > > >
> > > > I appreciate the additional clarification, and am raising my score to a 7. I suggest the authors incorporate the new experiments into the next version of the manuscript as they significantly strengthen the paper.

---

> > > > > ### Author Response · Authors · 2023-08-14
> > > > >
> > > > > Thank you for helping us clarify our experiments and for re-evaluating our paper! We plan on incorporating these new experiments into the next version.

---

### Official Review · Reviewer_K762 · 2023-06-27

**Soundness:** 3 good
**Presentation:** 3 good
**Contribution:** 3 good
**Rating:** 7
**Confidence:** 4

**Summary:**

The authors propose INVPROP, an algorithm for constructing certified bounds on the pre-image of neural networks with ReLU activations. The method constructs a convex over-approximation of the pre-image by implicitly optimizing the dual relaxation of a MILP problem. Empirical evaluation shows considerable improvement over the state of the art.

**Strengths:**

The authors' method is elegant and clear. Furthermore, I can see how it could be used as the baseline for many future works. Inverting neural networks is a notoriously difficult problem, and I commend the authors for proposing a workable solution.

**Weaknesses:**

The paper is dense to read at times, and the connection to existing works is often left to the reader. More specifically, most of the related work is presented at the end of the paper (Section 6). The state-of-the-art methods in neural network inversion (Rober 2022, Everett 2022) are mentioned for the first time in lines 249 and 256. Furthermore, no mention of earlier works in neural network inversion (see examples below) leave the reader with the feeling that this is a new problem that was never studied before:

	Kindermann, Linden, 1990, "Inversion of neural networks by gradient descent"
	Saad, Wunsch, 2007, "Neural network explanation using inversion"

I think the authors' contribution would shine even more if they introduced a little more historical perspective on the topic.

While reading the paper, I found a few minor writing issues, which I list below. For more detailed algorithmic question, see the "questions" and "limitations" secvtions of this review.

Many unnumbered equations make referencing difficult. See for instance lines 89, 144 and 166 in the main paper, lines 525-526 in Appendix E.

Lines 50-53. This contribution is difficult to parse without a strong understanding of ab-CROWN and local robustness problems.

Robustness verification is presented in Section 5. Given the structure of the paper, I believe it should be in Section 4.3.

Line 314, "forumalation" should be "formulation".

Appendix E, the output domain weights H and d are not present in the proof, which I guess it is for clarity, but the authors do not acknowledge this.

Appendix E, in lines 525-526 there is an extra term \sum c^ix^i in the objective that is not explained.

The formatting of figures and tables in Section 4 is ugly. At least, all tables and figures in page 8 should be flushed to the top, in order to avoid having a piece of main text in between them.


**Questions:**

Line 77. Why is inverting a neural network an intractable problem? It is clearly possible to compute it, albeit expensive to do so. In the end, f is a Lipschitz-continuous function, thus locally invertible.

After line 89, why is the first matrix not the identity? Surely the robot cannot teleport on the x_1 dimension. Also, why can the policy only control diagonal movement along the line [0.5, 1]? If these numbers are chosen just for illustration
purpose, then why presenting the "robot in a 2d environment" story altogether?

How are the concrete bounds [l,u] computed? Algorithm 1 says "cheap bound propagation", but these are only applicable for the forward verification problem. how to compute them for a general problem?

At Line 166, the proposed bounds yield a parallelogram domain around ReLU(x), whose shape is regulated by z. Why not using the classic, traingular domain from DeepPoly? What are the advantages of your parallelogram domain here?

**Limitations:**

The authors are not up-front on the fact that c in Equation (1) is extracted at random. The problem of how to do so is finally discussed in Sections 4 and 7. However, it is not clear how the choice of c impact the tightness of the pre-image. More specifically, in line 247, sampling 40 random planes may be ok for this particular 2-dimensional problem. Similarly, I can see how this works well for the OOD task in Section 4.2. At the same time, it is unclear how well that works for the MNIST task in Section 5. The authors offer no comment about this therein.

---

> ### Author Rebuttal · Authors · 2023-08-10
>
> Thank you for the helpful feedback! Thanks to your review, we have run more expansive evaluations (including **8112-dimensional inputs** and **167k neurons**), improved our discussion of limitations, and better situated ourselves within prior work.
>
> > I think the authors' contribution would shine even more if they introduced a little more historical perspective on the topic.
>
> We agree with the suggestion of better situating this approach within the context of prior work. We have added references to neural network inversion and discussed our relationship with this work.
>
> > Line 77. Why is inverting a neural network an intractable problem? It is clearly possible to compute it, albeit expensive to do so. In the end, f is a Lipschitz-continuous function, thus locally invertible.
>
> Thank you for pointing this out, we tone down our claim to discuss how it is computationally difficult to compute the exact pre-image. Our original writing was discussing the fact that neural networks are not strictly invertible, but we agree that they are locally invertible.
>
> > After line 89, why is the first matrix not the identity? Surely the robot cannot teleport on the x_1 dimension. Also, why can the policy only control diagonal movement along the line [0.5, 1]? If these numbers are chosen just for illustration purpose, then why presenting the "robot in a 2d environment" story altogether?
>
> We agree that the robot scenario does not properly highlight our story. We change this story to the DoubleIntegrator task we target later in the paper (since these dynamics are precisely the ones we verify in Section 4.1).
>
> > How are the concrete bounds [l,u] computed? Algorithm 1 says "cheap bound propagation", but these are only applicable for the forward verification problem. how to compute them for a general problem?
>
> We supply our initial bounding box for the input domain, denoted $l^{(0)}, u^{(0)}$, to a forward verification algorithm as an input set to over-approximate. This allows us to initialize [l, u] for every layer with a loose over-approximation.
>
> >At Line 166, the proposed bounds yield a parallelogram domain around ReLU(x), whose shape is regulated by z. Why not using the classic, traingular domain from DeepPoly? What are the advantages of your parallelogram domain here?
>
> The parallelogram domain is equivalent to the classic triangle domain from DeepPoly. This is simply a reparameterization of those bounds for mathematical convenience and compatibility with the GCP-CROWN paper (Zhang et al., 2023). When z is projected out, the relaxation becomes triangular. Using triangular bounds would cause no change in tightness and would only alter the formulation of our theorem.
>
> > The authors are not up-front on the fact that c in Equation (1) is extracted at random
>
> > it is not clear how the choice of c impact the tightness of the pre-image.
>
> In our control and OOD example, we utilize 40 evenly spaced hyperplanes as mentioned in Line 247. The selection of hyperplanes is a domain-specific consideration and is a separate consideration from the INVPROP algorithm. In higher dimensional applications, using two planes per input dimension, the positive and negative unit vectors, gives a box over-approximation of the pre-image. We find that this scales well in practice for robustness verification tasks (as it offers improvements for robustness verification in dimension 8112 and control verification in dimension 6) and scales well in theory since the number of hyperplanes necessary would be linear in the input dimension.
>
> —
>
> We appreciate the detailed analysis of our writing and agree with all of the minor writing issues. All of these issues will be addressed in a camera-ready version of this paper. We hope that we’ve addressed the proposed concerns and would be more than happy to answer additional questions!

---

> > ### Comment · Reviewer_K762 · 2023-08-11
> >
> > Many thanks for your response to my comments, and the additional empirical work that you put to address them.

---

### Official Review · Reviewer_PXif · 2023-07-06

**Soundness:** 4 excellent
**Presentation:** 2 fair
**Contribution:** 3 good
**Rating:** 7
**Confidence:** 4

**Summary:**

The paper considers verified neural network inversion. Given a set of output values, the goal is to compute the pre-image of these, i.e., the inputs that lie within this output set (given as a linear constraint over the output space).
To address this, the authors propose INVPROP, an algorithm that computes
an over-approximation of the convex hull of the pre-image.

This set is represented as the intersection of multiple half-space constraints.

Given a direction vector finding the corresponding half-space constraint can be phrased as an optimization problem. Formulating this as a mixed-integer linear program (MILP) allows to recover a tight convex hull (assuming enough direction vectors are used). To improve efficiency at the cost of tightness this MILP can be relaxed into a LP. This is similar to the same problem in standard forward verification, however, in contrast to there, a constructive closed-form solution to the LP is not available.

Inspired by the CROWN verification algorithm the authors propose to consider the dual of the LP. This formulation allows to obtain a lower bound of the LP and thus a over-approximation to the convex hull. This approach, termed INVPROP, now can be solved by traversing the neural network back-to-front and obtaining neuron bounds via gradient ascent.
As in forward verification, this can be tightened by branching the problem and obtaining bounds for sub-problems.

INVPROP is evaluated: on backward-reachability for neural control policies, out-of-distribution detection, and robustness verification (combined with a forward verification algorithm), where it achieves strong results.


**Strengths:**

- Creative and Novel Approach
- The mathematical results appear sound and well motivated
- Good empirical results, though -- other than the improvements in certified robustness -- on niche benchmarks and settings
- Well connected with related and prior work

**Weaknesses:**

- While overall quite easy to read/follow, the presentation appears a bit sloppy and rushed
- There is no discussion of the limitations and scalability of the approach, other than the mention of per-neuron branching
- While the evaluation highlights why the approach is useful, it does not provide a good understanding of the limitations of the approach, i.e. the first two tasks are rather low-dimensional and the verification task is only discussed very briefly

See the below questions for more details and addressable points.

**Questions:**

- What is $z$ in (2a)? Is it a typo and not needed for the optimization problem?
- Line 267 refers to Figure 4 which is just found in the appendix.  Even with the Figure the description in line 267/268 is not clear to me. Can you clarify the setting in the OOD approach?
- Can you discuss the scalability of the approach? What are the key limiting factors? How large is the role of the input dimensionality? Ideally, this is also shown through further experiments, but I understand that this is hard within the constraints of the rebuttal.


EDIT POST REBUTTAL: My questions have been answered, thereby addressing the above weaknesses. I have thus updated my score.

**Limitations:**

There is some discussion of limitations though it is not very clear to me what the key limiting factors, e.g. on the scalability of the approach are. (See questions.)
Societal impact is discussed in a manner appropriate to the paper.

---

> ### Author Rebuttal · Authors · 2023-08-10
>
> Thank you for the helpful feedback! Thanks to your review, we have run more expansive evaluations (including **8112-dimensional inputs** and **167k neurons**), extended our reachability analysis to a **6-dimensional state quadrotor**, and improved our discussion of limitations.
>
> > Can you discuss the scalability of the approach? What are the key limiting factors? How large is the role of the input dimensionality? Ideally, this is also shown through further experiments, but I understand that this is hard within the constraints of the rebuttal.
>
> > While the evaluation highlights why the approach is useful, it does not provide a good understanding of the limitations of the approach, i.e. the first two tasks are rather low-dimensional and the verification task is only discussed very briefly
>
> In light of these suggestions, we have run new experiments to highlight the strength of our method. In the verified robustness setting, our method handles 8112-dimensional inputs for 167k neuron networks. In the control setting, our method handles a 6-dimensional quadrotor control problem. Furthermore, we provide more details on scalability across dimensions/neurons/branching in our new limitations section. Please see our new results and limitations in our general response and PDF.
>
> > There is no discussion of the limitations and scalability of the approach, other than the mention of per-neuron branching
>
> We agree with the reviewer that we have not adequately discussed our limitations with respect to how INVPROP as well as node branching scales with size. We have rewritten a more comprehensive limitations section, which we provide in the general response. We hope this addresses the concerns raised in this regard.
>
> > What is z in (2a)? Is it a typo and not needed for the optimization problem?
>
> Thank you for spotting this typo, we have fixed it.
>
> > Line 267 refers to Figure 4 which is just found in the appendix. Even with the Figure the description in line 267/268 is not clear to me. Can you clarify the setting in the OOD approach?
>
> We agree that we have not properly introduced the OOD setting, and we add the following section to our Appendix to properly describe the exact setup in this case.
>
> A simple approach to train a classifier that “knows when it doesn’t know” is to train a classifier with an additional logit representing (out of distribution) OOD-ness. Labeled data for this additional class is generated via outlier exposure, i.e., adding synthetic training data with features drawn from an OOD distribution (known as outlier exposure [1]) and labels corresponding to the additional OOD logit [2]).
>
> Consider the logits $y$ produced by a classifier trained in this manner for a binary classification task. The classifier can classify in-distribution data by comparing $y_0$ and $y_1$, the logits corresponding to the two in-distribution labels. The quantity $\max(y_0, y_1) - y_2$, known as the logit gap, can be used to identify out-of-distribution data (Figures 3 and 4).
>
> A natural question to ask is: Can it correctly identify data points far from the training data as being OOD? In order to do so, one needs to compute the preimage of the set $\mathcal{S}_{\text{out}} = \{y : \max(y_0, y_1) \geq y_2\}$. We detail how to model this set with linear constraints in Appendix F.1. INVPROP enables us to compute an overapproximation of the pre-image, thereby answering the question of whether the classifier learned to correctly identify OOD data.
>
> In the OOD example, the model was trained to distinguish three classes. Two are defined by the two classes in Figures 3 and 4. The third class represents “out-of-distribution” inputs. All inputs outside of the two circles should be assigned to the OOD class.
>
> > While overall quite easy to read/follow, the presentation appears a bit sloppy and rushed
>
> Thank you for your feedback in this regard! We would love to improve our presentation and would appreciate further concrete suggestions on what sections could use the most help.
>
> —
> We hope that we’ve addressed the proposed concerns and would be more than happy to answer additional questions!
>
> [1] Deep anomaly detection with outlier exposure, D Hendrycks, M Mazeika, T Dietterich
> [2] ATOM: Robustifying Out-of-distribution Detection Using Outlier Mining, Jiefeng Chen, Yixuan Li, Xi Wu, Yingyu Liang, Somesh Jha

---

> > ### Comment · Reviewer_PXif · 2023-08-11
> > **Reply**
> >
> > I thank the authors for their response.
> > The rebuttal indeed addresses most of my concerns and I'm particularly delighted by the scope of the additional experiments.
> > I currently do not have further questions and have updated my score in the light of the new results and explanations.

---

### Official Review · Reviewer_bSxe · 2023-07-06

**Soundness:** 3 good
**Presentation:** 3 good
**Contribution:** 3 good
**Rating:** 7
**Confidence:** 4

**Summary:**

The paper considers the problem of computing approximations of the set of
inputs for which a given neural network satisfies certain output constraints.
To solve the problem, the paper discusses adaptations of a number of approaches
from standard neural network verification, including MILP and LP formulations,
and motivates the development of bound propagation methods and optimisations of
the Lagrangian duals. Experimental results of the resulting method are reported
on (i) backward reachability analysis for neural feedback loops; (ii) OOD
Detection; (iii) robustness verification.

**Strengths:**

The problem studied is well motivated and its applicability is sufficiently
exemplified.

The adaptation of the α,β-Crown method to solve the problem is novel.

The efficacy of the resulting method is evaluated on a range of benchmarks,
where it is shown to outperform the state-of-the-art in backward reachability
analysis. The connection of backward propagation to standard forward
verification is brilliant to see, especially its utilisation towards more
efficient forward verification.

**Weaknesses:**

The contribution is incremental to the α,β-Crown method.

At its present stage the method is limited to low-dimensional networks. Also,
no scalability is discussed with respect to big (but still low-dimensional)
networks.

An important feature of the α,β-Crown framework on which the method is based,
namely node branching, is not discussed but left as future work, thereby giving
a "preliminary" feel to the paper.


**Questions:**

Could you please discuss the overhead of the incorporation of the output
constraint to forward verification when larger, i.e., with millions of ReLU
nodes, networks are considered?


**Limitations:**

Adequately addressed.

---

> ### Author Rebuttal · Authors · 2023-08-10
>
> Thank you for the helpful feedback! Thanks to your review, we have run more expansive evaluations (including **8112-dimensional inputs** and **167k neurons**), **fully implemented branching** from abCROWN with INVPROP, and improved our discussion of limitations.
>
> > At its present stage the method is limited to low-dimensional networks. Also, no scalability is discussed with respect to big (but still low-dimensional) networks.
>
> In light of these suggestions, we have run new experiments to highlight the strength of our method. In the verified robustness setting, our method handles 8112-dimensional inputs for 167k neuron networks. In the control setting, our method handles a 6-dimensional quadrotor control problem. Furthermore, we provide more details on scalability across dimensions/neurons/branching in our new limitations section. Please see our new results and limitations in our general response and PDF.
>
> > The contribution is incremental to the α,β-Crown method.
>
> Though INVPROP builds on the alpha-beta-CROWN framework, we believe that this work represents many significant technical contributions such as iteratively refining intermediate bounds, incorporating constraints sequentially after a layer’s intermediate bounds, and leveraging constraints on the output layer. None of these were supported by existing alpha-beta-CROWN, and INVPROP enables tight verification for novel applications such as control, and tightens bounds for existing robustness verification problems.  Since we’re compatible with the framework, we can easily leverage existing state-of-the-art verification tools, as shown in our new adversarial robustness verification experiments.
>
> > An important feature of the α,β-Crown framework on which the method is based, namely node branching, is not discussed but left as future work, thereby giving a "preliminary" feel to the paper.
>
> In our new experiments, we have integrated INVPROP with alpha-beta-CROWN and **fully implemented branching** over neurons and inputs. Utilizing Branch and Bound for verification is independent of this work, and the trade-off between the tightness of verification and computational constraints is well-studied in prior literature.
>
> > Could you please discuss the overhead of the incorporation of the output constraint to forward verification when larger, i.e., with millions of ReLU nodes, networks are considered?
>
> In our new limitations section, we discuss how the output constraint scales with neural network size. Specifically, using the output constraints for tightening the intermediate bounds for every layer incurs a quadratic dependence in runtime, but we can use our framework for tightening a single layer and still see the benefits, as seen in our new evaluation results for the YOLO benchmark in VNN-COMP 2023.
>
> —
>
> We hope that we’ve addressed the proposed concerns and would be more than happy to answer additional questions!

---

> > ### Comment · Reviewer_bSxe · 2023-08-16
> > **Thank you for the response**
> >
> > Thank you for the detailed response which addresses the minor concerns of my review. I have thus increased my score from 6 to 7.

---

### Author Rebuttal · Authors · 2023-08-10

We thank the reviewers for their helpful and constructive comments about our paper.

Thanks to the reviews, we have run a large suite of new experiments where we have evaluated our method on robustness verification for **inputs of dimension 8112** and neural networks of **167k neurons**, including complicated networks which feature convolutional layers, skip connections, and pooling. We also introduce new experiments for over-approximating the pre-image in a **6-dimensional quadrotor control** problem, where we improve upon state-of-the-art by **257 times tighter, 3.29 times faster**. We believe this addresses many concerns about the practical scalability of INVPROP, hyperplane selection, and branching. Furthermore, we provide a more nuanced discussion of these components of our approach in our new limitations section.

**New robustness verification experiments on large networks**

We performed an additional experiment using a benchmark from the VNN-COMP 2023. We extended alpha-beta-CROWN with support for output constraints using INVPROP. To reduce the runtime cost of incorporating output constraints, they were only applied to the penultimate layer of the network. Also, they are only used during the alpha-CROWN computation, i.e., during the optimization of relaxations of intermediate layers. If this step fails to prove UNSAT, alpha-beta-CROWN performs branching over the hidden neurons to remove nonlinearities. As alpha-beta-CROWN doesn’t update the intermediate relaxations in this second step, output constraints are not applied, however, the tighter intermediate layer bounds obtained from INVPROP in the first step (alpha-CROWN) also help improve the bounds after branching. We provide a comparison of the original alpha-beta-CROWN and our extended version in Figure 1. With output constraints, 5 of the 9 instances that **previously timed out (300s) were now solved on average within 16.2 seconds**. For instances that can be verified by vanilla alpha-beta-CROWN, INVPROP reduces the verification time on average 13.9 seconds.

**New quadrotor benchmark for higher dimension control**

We consider over-approximating the pre-image of the quadrotor discussed in Figure 15 of [1]. This quadrotor has a 6-dimensional state space, with 3 dimensions dedicated to position in the (x,y,z) coordinate space. We run prior work and our algorithm to find the tightest bounding box in this 3d space, reflecting a total of 6 constraints (one for each direction in each dimension). We run our algorithm with no branching. To benchmark against prior work, we run ReBreach-LP with no output partitioning and ReBreach-LP with 15625 output partitions using SkipLP, disciplined parametric programming, and the guided partitioning algorithm discussed in [1]. In Figure 2 and Table 1, we compare these three strategies, and we find that our over-approximation of the pre-image (in blue) is smaller than the state-of-the-art over-approximations by **257 times, 3.29 times faster**. Therefore, our algorithm scales well to higher dimensional control examples.

**New limitations section**

For higher-dimensional instances, our method would best work for box over-approximations (which require $2d$ hyperplanes in $d$ dimensions) and would struggle at more complicated shapes such as sphere pre-images. Though our hyperplane selection strategy works well in our application domains, we expect that more complex problems will benefit from a domain-specific strategy.

Iteratively refining all of the intermediate bounds utilizing INVPROP faces a quadratic dependence on network depth, as opposed to a linear dependence in traditional forward verification. Otherwise, our method scales similarly to verification of adversarial robustness [3] as can be seen in “Connection to forward verification”. To mitigate this, output constraints could only be applied to layers close to the output, as we used for the VNN-COMP23 YOLO benchmark.

Though our branching strategy works for verification in our case, it might not scale to higher dimensions. However, utilizing Branch and Bound for verification is independent of this work, and the trade-off between the tightness of verification and computational constraints is well-studied in prior literature ([2], [3], [4]).

—

We hope that we have addressed any concerns and would be happy to answer any further questions to help inform an accurate assessment of our contribution!

[1] Backward Reachability Analysis of Neural Feedback Loops: Techniques for Linear and Nonlinear Systems, Nicholas Rober, Sydney M. Katz, Chelsea Sidrane, Esen Yel, Michael Everett, Mykel J. Kochenderfer, Jonathan P. How
[2] Branch and Bound for Piecewise Linear Neural Network Verification, Rudy Bunel, Jingyue Lu, Ilker Turkaslan, Philip H.S. Torr, Pushmeet Kohli, M. Pawan Kumar
[3] Beta-CROWN: Efficient Bound Propagation with Per-neuron Split Constraints for Complete and Incomplete Neural Network Robustness Verification, Shiqi Wang, Huan Zhang, Kaidi Xu, Xue Lin, Suman Jana, Cho-Jui Hsieh, J. Zico Kolter
[4] IBP Regularization for Verified Adversarial Robustness via Branch-and-Bound, Alessandro De Palma, Rudy Bunel, Krishnamurthy Dvijotham, M. Pawan Kumar, Robert Stanforth

---

### Decision · Program_Chairs · 2023-09-21

**Decision:**

Accept (spotlight)

**Comment:**

The reviewers are happy to accept the paper. Please incorporate all the discussions and new experiments into the final version.